# INTERACTIVE IMPLICIT IN-CONTEXT LEARNING

## ABSTRACT

In-context learning (ICL) empowers large language models (LLMs) to generalize from few-shot demonstrations but faces challenges with quadratic computational complexity and unstable performance as demonstration counts increase. Implicit ICL ($I^2CL$) addresses these limitations by encoding demonstrations into a unified task vector injected during inference, achieving significant efficiency gains and order invariance. However, existing approaches prioritize inference efficiency, sacrificing interactive capabilities crucial to standard ICL—inter-demonstration and demonstration-query interactions. Substantial research underscores that these interactions are fundamental for strong reasoning performance. To bridge this critical gap, we propose Interactive Implicit In-Context Learning ($I^3CL$), a simple yet effective framework that restores such essential capabilities of standard ICL. $I^3CL$ integrates only two lightweight interactive modules, preserving $I^2CL$'s core efficiency with minimal computational overhead. Experimental evaluation across nine diverse tasks using four LLMs show that $I^3CL$ delivers a significant average performance gain of 7.26% over prior $I^2CL$ baselines. This substantial improvement strongly indicates that restoring interactive capabilities is essential for advancing the effectiveness of implicit ICL methods.

## 1 INTRODUCTION

In-context learning (ICL) (Brown et al., 2020) enables large language models (LLMs) to learn tasks using only a few input examples, eliminating the need for parameter updates or explicit fine-tuning. This approach has become a key technique for efficient LLM deployment (Luo et al., 2024). Despite promising results across diverse reasoning tasks (Li et al., 2023; Liu et al.; Zhang et al., 2025), ICL suffers from significant limitations: sensitivity to demonstration format and order (Zhao et al., 2021; Lu et al., 2022), and quadratic computational and memory overhead as demonstration count increases (Wang et al., 2023; Li et al., 2025). These constraints hinder real-world scalability.

To address these limitations, recent studies have turned to the implicit ICL paradigm (Hendel, 2023; Wang et al., 2023; Liu et al., 2024; Li et al., 2025). Unlike standard ICL, which appends the token sequences of all demonstrations into the input prompt, implicit ICL encodes each demonstration into a dense vector. These vectors are then aggregated into a compressed vector, called a "task vector". During inference, this task vector is injected directly into the LLM's latent space to steer the generation process. This approach delivers key advantages: (1) It introduces minimal computational overhead during inference (only vector operations), achieving nearly zero-shot inference speeds. (2) By merging demonstrations via order-irrelevant aggregation, it eliminates the order sensitivity inherent in standard ICL.

While showcasing promising performance, current implicit ICL methods prioritize inference speed at the expense of the core mechanisms underpinning ICL's effectiveness. This limitation manifests in two key aspects, as exemplified by the current state-of-the-art (SOTA) implicit ICL method $I^2CL$ (Li et al., 2025). First, during demonstration encoding, $I^2CL$ processes demonstrations independently, isolating each into a vector. This contrasts with standard ICL where self-attention mechanisms explicitly model interactions between demonstrations. These interactions are crucial for constructing a powerful task vector that adapts LLMs to new tasks (Hendel, 2023; Saglam et al.). Extensive research confirms that such inter-demonstration interactions (e.g., dynamic information flow during encoding) are fundamental to models forming strong reasoning capabilities (Xie et al., 2022; Yasunaga et al., 2023). By neglecting these interactions, $I^2CL$ risks limiting models to recogniz-

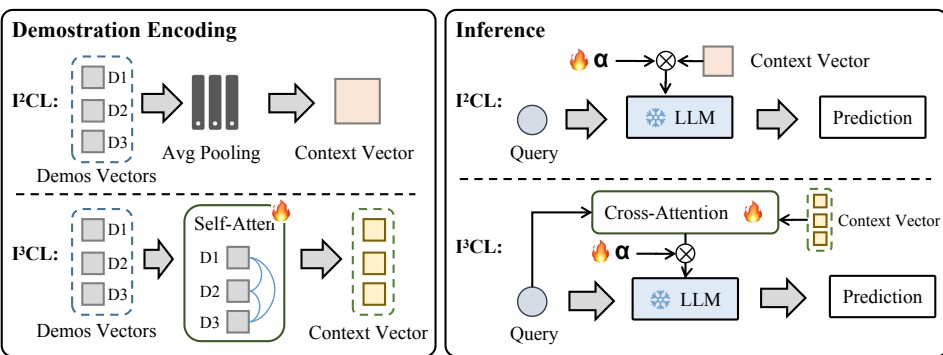

Figure 1: Key differences between I$^2$CL and our I$^3$CL.

ing only surface patterns within isolated demonstrations, constraining their capacity to synthesize higher-order patterns necessary for sophisticated reasoning.

Second, during task vector distillation, I$^2$CL relies on simple average pooling of the encoded demonstration vectors. Despite high efficiency, this naive averaging strategy treats all demonstrations as equally informative. Theoretically, this leads to high entropy loss, flattening the uneven information distribution across demonstrations. Consequently, the model is hindered from leveraging the varying relevance of demonstrations to the test query. In contrast, standard ICL employs attention mechanisms that dynamically reweight demonstrations based on the test query, enabling the model to focus on the most relevant demonstrations (Yousefi et al., 2023; Chen et al., 2024). This selective attention mechanism is essential for preserving the highest-value information for reasoning.

These limitations prevent current implicit ICL methods from harnessing the fundamental capabilities of standard ICL: effectively combining information across demonstrations and selecting the most relevant demonstrations. To address this gap, we introduce Interactive Implicit In-Context Learning (I$^3$CL). I$^3$CL instills these crucial capabilities via two simple yet effective interaction modules designed to mirror the core strengths of standard ICL. Figure 1 highlights the key differences between I$^3$CL and I$^2$CL. I$^3$CL introduces two critical levels of interaction: (*i*) **Inter-demonstration interaction (encoding)**: Demonstrations exchange information to form an expressive task vector; and (*ii*) **Demonstration-to-query interaction (inference)**: The task vector is conditionally modulated by the query before injection, enabling the model to emphasize the most relevant demonstrations. Both interactions are built through lightweight attention mechanisms, preserving inference speeds close to I$^2$CL. While conceptually simple, I$^3$CL represents a significant conceptual departure: shifting the implicit ICL paradigm from "sacrificing performance for efficiency" towards "recovering essential capabilities while preserving efficiency", effectively addressing fundamental weaknesses in current implicit ICL methods. This shift delivers substantial performance improvements with minimal computational overhead.

We evaluate I$^3$CL across four LLMs and nine tasks. Results confirm strong generalizability of our approach across diverse tasks and model architectures. Crucially, I$^3$CL consistently outperforms prior implicit ICL baselines, achieving an average accuracy improvement of 7.26%. Comprehensive analyses, including ablation studies, scalability testing, efficiency profiling and task vector comparisons, proves the superiority of our approach.

## 2    RELATED WORK

As LLMs scale, the computational cost of full fine-tuning rises dramatically. ICL provides a more resource-efficient alternative, enabling LLMs to adapt to new tasks using only a few demonstrations without additional training (Luo et al.). However, ICL remains unstable, exhibiting high sensitivity to the content and format of these demonstrations. Furthermore, its computational and memory requirements increase quadratically with the sequence length, making larger demonstration sets particularly costly. Current research addressing these limitations primarily focuses on three lines: prompt-based methods, implicit ICL methods and prompt compression.

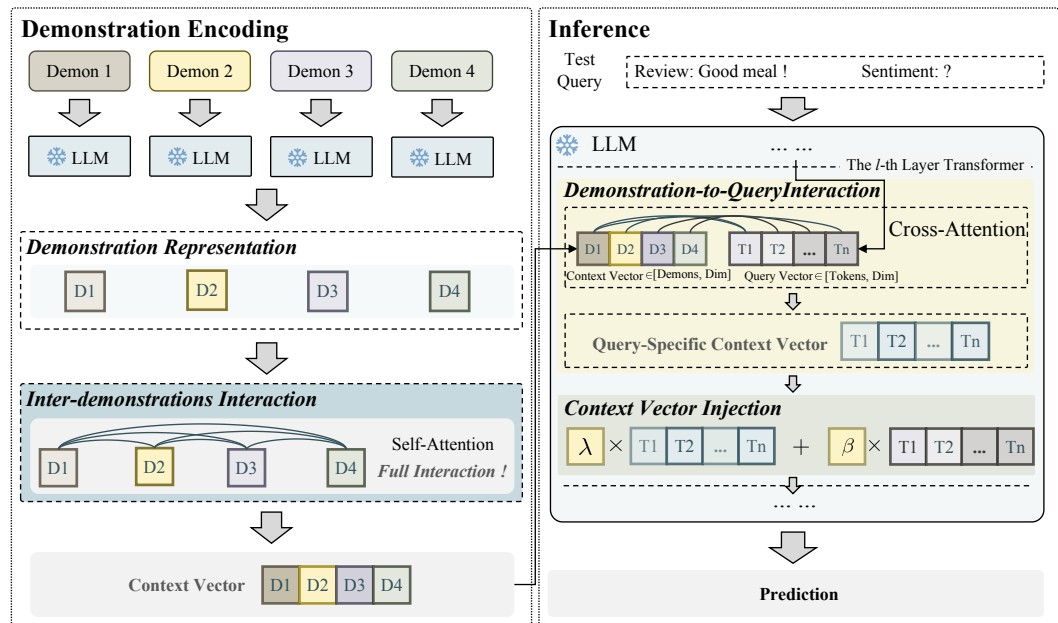

Figure 2: Overview of our Interactive Implicit In-context Learning I[3]CL framework.

**Prompt-based Methods** These methods aim to analyze how demonstration factors influence ICL performance, guiding the design of more effective prompt templates and selection strategies (Rubin et al., 2022; Yin et al., 2023; Wan et al., 2023; Fan et al., 2024; Peng et al., 2024; Jin et al., 2025). For instance, Rubin et al.; Yin et al. focused on optimizing and retrieving prompt templates to reduce the cost of manual prompt engineering. Fan et al. used contrastive examples based on the minimum-edit principle to address information bias, while Peng et al. proposed "TopK+ConE," a selection strategy combining data features and model behavior.

**Implicit ICL Methods** The core of these methods is to extract a unified "task vector" from a few demonstrations, and inject it back into the LLM to guide reasoning of test samples (Hendel, 2023; Wang et al., 2023; Liu et al., 2024; Li et al., 2025). For example, Hendel interpreted ICL as compressing demonstrations into a task vector that conditions transformer behavior, providing a novel theoretical perspective. Liu et al. introduced a context-vector framework for open-ended generation that encodes task information into a single vector, reducing prompt length and improving efficiency. More recently, Li et al. introduced I[2]CL, which injects task vectors into the residual stream through learnable parameters, achieving promising performance on the few-shot tasks.

**Prompt Compression Methods** Prompt compression methods (Chang et al., 2024) share the objective of improving inference efficiency in LLMs through mitigating the redundant context in the prompt (Chevalier et al., 2023; Ge et al., 2023; Yen et al., 2024). For instance, AutoCompressors (Chevalier et al., 2023) recursively compressed long contexts into summary vectors, which are injected as soft prompts to extend context capacity. CEPE (Yen et al., 2024) employed a lightweight encoder and cross-attention to process long texts in chunks, feeding them into a frozen decoder-only model to extend context length with minimal overhead.

While building upon the implicit ICL framework, I[3]CL differs in two key aspects: (1) Previous methods typically treat demonstrations as independent samples, encoding each in isolation. This independent processing limits the model's ability to exchange information between demonstrations—a core aspect of standard ICL. Our approach overcomes this by incorporating an inter-demonstration interaction to explicitly model these dependencies. (2) Prior techniques often aggregate demonstration vectors via heuristics like average pooling. However, demonstration selection is known to be crucial (Wang et al., a; Liu et al., 2022). Such naive aggregation tends to induce high-entropy loss, flattening the uneven information distribution across demonstrations. In contrast, our method reinstates this critical selection mechanism inherent in standard ICL via demonstration-to-query

interaction. Together, these complementary interaction mechanisms enable I$^3$CL to generate more compressive and discriminative task vectors for in-context learning.

# 3 METHODOLOGY

We propose Interactive Implicit In-Context Learning (I$^3$CL), enhancing I$^2$CL (Li et al., 2025) with two core components: (*i*) an **inter-demonstration interaction** module that enables demonstrations to exchange information; and (*ii*) a **demonstration-to-query interaction** module that dynamically weights demonstrations based on the test query. We first formalize the ICL task objective, then detail our approach in subsequent sections. The overview of I$^3$CL is presented in Figure 2.

## 3.1 TASK DEFINITION FOR ICL

Given a task, ICL assumes a demonstration set $\mathcal{D} = \{(x_1, y_1), (x_2, y_2), \ldots, (x_N, y_N)\}$, where each pair $(x_i, y_i)$ consists of an input $x_i$ and its corresponding label $y_i$. During inference, the task instruction $I$, the demonstration set $\mathcal{D}$, and the test query $x_t$ are combined to form a prompt $\mathcal{P}_t = \{I, \mathcal{D}, x_t\}$, which is fed into the LLM to generate the prediction $\hat{y}_t$ from the label space $\mathcal{Y}$.

## 3.2 INTER-DEMONSTRATION INTERACTION

To encode each demonstration $(x_i, y_i) \in \mathcal{D}$, we first form $\mathcal{I}_i = \{I, x_i, y_i\}$ by concatenating the task instruction $I$, input $x_i$ and label $y_i$. $\mathcal{I}_i$ is then passed through the LLM, and we extract the hidden representations from both the multi-head attention (MHA) module and the feedforward network (MLP) at each layer of the transformer model $\mathcal{M}$:

$$\langle a_i^l, m_i^l \rangle = \mathcal{M}_\theta^l(\mathcal{I}_i), \tag{1}$$

where $l$ indexes the transformer layer; $a_i^l \in \mathbb{R}^{s \times d}$ denotes the MHA module output for layer $l$; $m_i^l \in \mathbb{R}^{s \times d}$ denotes the MLP output for layer $l$; $s$ and $d$ are the sequence length and representation dimension, respectively.

Prior approaches (Liu et al., 2024; Li et al., 2025) typically represent each demonstration using only the Label Last Token (LLT), i.e., the final token in sequence $\mathcal{I}_i$. However, causal language models (CLMs) exhibit positional bias that causes the LLT's attention distribution to concentrate excessively on its own position under autoregressive objectives (Wang et al., b). This self-referential bias limits the LLT's capacity to integrate information from task instructions ($I$) and input contexts ($x_i$). To mitigate this, we introduce the Prompt Last Token (PLT), defined as the last token of the $\{I, x_i\}$, as a complementary signal. We construct a compressive demonstration representation by averaging the LLT and PLT embeddings at each layer $l$:

$$\bar{a}_i^l = \frac{1}{2}(\bar{a}_{i,\text{PLT}}^l + \bar{a}_{i,\text{LLT}}^l) \in \mathbb{R}^d; \bar{m}_i^l = \frac{1}{2}(\bar{m}_{i,\text{PLT}}^l + \bar{m}_{i,\text{LLT}}^l) \in \mathbb{R}^d, \tag{2}$$

where $\bar{a}_{i,\text{PLT}}^l \in \mathbb{R}^d$ and $\bar{m}_{i,\text{PLT}}^l \in \mathbb{R}^d$: PLT embeddings from MHA/MLP outputs; $\bar{a}_{i,\text{LLT}}^l \in \mathbb{R}^d$ and $\bar{m}_{i,\text{LLT}}^l \in \mathbb{R}^d$: LLT embeddings from MHA/MLP outputs.

We concatenate $\bar{a}_i^l$ and $\bar{m}_i^l$ to form layer-$l$ context vector for the $i$-th demonstration $c_i^l = [\bar{a}_i^l; \bar{m}_i^l] \in \mathbb{R}^{2d}$. Stacking all demonstration vectors yields the demonstration set's layer-$l$ context matrix:

$$C^l = [c_1^l; c_2^l; \cdots; c_{|D|}^l] \in \mathbb{R}^{|D| \times 2d}. \tag{3}$$

Subsequently, we design a lightweight interaction module based on the self-attention mechanism to model the dependencies among demonstrations. Specifically, we adopt the standard scaled dot-product self-attention Vaswani et al. (2017) with trainable parameters. For the $l$-th layer's context vector $C^l \in \mathbb{R}^{|D| \times 2d}$, we project it into Query (Q), Key (K), and Value (V) vectors using trainable weight matrices $W_Q, W_K, W_V \in \mathbb{R}^{2d \times 2d}$. The output is then computed as:

$$C'^l = \text{Softmax}\left(\frac{QK^T}{\sqrt{d}}\right)V, \tag{4}$$

where $Q = C^l W_Q$, $K = C^l W_K$, and $V = C^l W_V$.

The motivation for inter-demonstration interaction stems from the observation that task regularities cannot be fully revealed by a single demonstration; instead, they emerge from the joint implications of multiple demonstrations (Xie et al., 2022; Akyürek et al., 2022). When demonstrations are encoded independently, the model tends to capture only local individual regularities, but fails to establish global collective regularities, i.e., a compressive and discriminative task vector. A detailed analysis of this module is provided in section of further analysis and Appendix A.7.

### 3.3 DEMONSTRATION-TO-QUERY INTERACTION

To this end, we employ a cross-attention mechanism to assess the importance of each demonstration $(x_i, y_i) \in \mathcal{D}$ to the test query $x_t$. Specifically, taking the output of the $l$-th layer as example, the importance scores from MHA and MLP are computed as follows:

$$\text{score}^l_{a,x_t} = \text{softmax}(a^l_{x_t} \cdot c'^{l\top}_{\text{a}}); \ \text{score}^l_{m,x_t} = \text{softmax}(m^l_{x_t} \cdot c'^{l\top}_{\text{m}}), \tag{5}$$

where matrices $c'^l_a, c'^l_m \in \mathbb{R}^{|\mathcal{D}| \times d}$ represent the corresponding MHA and MLP parts extracted from the demonstration set's context matrix $C'^l$; $a^l_{x_t}, m^l_{x_t} \in \mathbb{R}^{s \times d}$ denote the MHA and MLP outputs of the test query $x_t$.

Overall, this equation derives normalized importance scores that quantify the relevance of each demonstration relative to each token in the test query. For example, $\text{score}^l_{a,x_t}$ is derived as follows: The dot product $a^l_{x_t} \cdot c'^{l\top}_a$ computes pairwise token-demonstration similarities. These raw similarities are normalized via softmax applied along the demonstration axis ($|\mathcal{D}|$), yielding the matrix $\text{score}^l_{a,x_t} \in \mathbb{R}^{s \times |\mathcal{D}|}$. Each element $\text{score}^l_{a,x_t}[i,j]$ quantifies the relevance of the $j$-th demonstration to the $i$-th token in $x_t$.

Based on these importance scores, we construct query-specific context vectors as follows:

$$c^{*\,l}_{a,x_t} = \text{score}^l_{a,x_t} \cdot c'^l_a; \ c^{*\,l}_{m,x_t} = \text{score}^l_{m,x_t} \cdot c'^l_m \tag{6}$$

where $c^{*\,l}_{a,x_t}$ and $c^{*\,l}_{m,x_t} \in \mathbb{R}^{s \times d}$ denote the query-specific context vectors for MHA and MLP, respectively.

These vectors are then injected as follows:

$$o^l_{x_t} = o^{l-1}_{x_t} + (\lambda^l_a \cdot c^{*\,l}_{a,x_t} + \beta^l_a \cdot a^l_{x_t}) + (\lambda^l_m \cdot c^{*\,l}_{m,x_t} + \beta^l_m \cdot m^l_{x_t}) \tag{7}$$

where $\lambda^l_a, \beta^l_a, \lambda^l_m, \beta^l_m$ are learnable coefficients weighting the query-specific context vectors and the original query vectors at the $l$-th layer; $o^l_{x_t} \in \mathbb{R}^{s \times d}$ is the layer's final output for the query, formed by adding the previous layer's output $o^{l-1}_{x_t}$ via a residual connection.

### 3.4 COEFFICIENT CALIBRATION

Scale mismatches between the fused query vectors and the LLM's original distribution can cause inference difficulties. To address this, the coefficient calibration serves to adapt the fused vectors for compatibility with the LLM's pre-trained parameters. Following I$^2$CL (Li et al., 2025), we employ a parameter-efficient fine-tuning (PEFT) strategy to optimize these coefficients. Specifically, we freeze all pre-trained parameters of the LLM and update only the following two sets of parameters via gradient descent: (1) the parameters of the interaction modules (e.g., the weight matrices $W_Q, W_K, W_V \in \mathbb{R}^{d \times d}$ in self-attention), and (2) the coefficients $\lambda$ and $\beta$. The optimization objective is to minimize the perplexity of the label tokens over the demonstration set $\mathcal{D}$, with the loss function defined as follows:

$$\mathcal{L} = -\frac{1}{|\mathcal{D}|} \sum_{(x,y) \in \mathcal{D}} \log P(y|x, \lambda, \beta, O^*_x) \tag{8}$$

where $P(y|x, \lambda, \beta, O^*_x)$ denotes the LLM's predictive probability of label $y$ for the test query $x$; $O^*_x$ denotes the last layer's output of the query; $\lambda$ and $\beta$ denotes the learnable coefficients.

This optimization constitutes an offline calibration process: the fusion coefficients $\lambda$ and $\beta$ are calibrated via this task-specific demonstration set $\mathcal{D}$ in a bootstrapping training stage. Once calibrated

through this procedure, these coefficients remain fixed during inference. Importantly, for tasks that require the model to generate multiple tokens, our approach injects contextual vectors at each decoding step. This ensures every generated token benefits from the in-context guidance, preserving coherence throughout long-form generation.

# 4 EXPERIMENTS

## 4.1 SETUP

**LLM Backbones**. We evaluate I³CL on four open-source LLMs: LLaMA2-7B (Touvron et al., 2023), LLaMA3-8B (Dubey et al., 2024), GLM4-9B (Zeng et al., 2024) and DeepSeek-R1 (Guo et al., 2025). These models are selected becuase of their prevalent use in prior implicit ICL works (Hendel, 2023; Liu et al., 2024; Li et al., 2025).

**Datasets**. We conduct a comprehensive evaluation of I³CL on eight diverse classification datasets, including sentiment classification: SST-2 (Socher et al., 2013), SST-5 (Socher et al., 2013); question classification: TREC (Voorhees & Tice, 2000); topic classification: AGNews (Zhang et al., 2015), DBPedia (Zhang et al., 2015); movie review classification: MR (Pang & Lee, 2005); subjectivity classification: Subj (PANG, 2004), and hate speech detection: HateSpeech18 (de Gibert et al., 2018). Besides, we expanded our evaluation to include a text generation task: text style transfer. Following the SOTA implicit ICV method (Li et al., 2025), we tested I³CL on the ParaDetox dataset (Logacheva et al., 2022), where the task is to rewrite toxic sentences into neutral expressions while preserving meaning. This requires precise context understanding and controlled generation capabilities.

**Compared Baselines**. The baseline methods are categorized into three groups: the basic LLM methods, the implicit in-context learning methods and the prompt compression methods. The first group includes the Zero-Shot method (Touvron et al., 2023), the conventional ICL method (Li et al., 2025), and the LoRA-based FineTuning method (LoRA-FT) (Shen et al., 2022). The second group focuses on prior implicit in-context learning methods, including Label-Anchor (Wang et al., 2023), Task-Vector (Hendel, 2023), ICV (Liu et al., 2024), and I²CL (Li et al., 2025). The last group includes AutoComp (Chevalier et al., 2023), ICAE (Ge et al., 2023) and CEPE (Yen et al., 2024). As I²CL applies random sampling to the test set without releasing the specific sampled instances, we replicate their sampling strategy to reconstruct the test set and reproduce all baselines based on this setup.

| Method | SST-2 | SST-5 | TREC | AGNews | HS18 | DBPedia | MR | Subj |
|---|---|---|---|---|---|---|---|---|
| Zero-Shot | 89.16 | 41.04 | 41.96 | 45.60 | 76.44 | 69.44 | 76.88 | 52.52 |
| ICL | 91.00 | 44.32 | 60.08 | 85.00 | 77.92 | 71.48 | 89.56 | 56.24 |
| LoRA-FT | 92.15 | 48.44 | 85.36 | 86.04 | 82.48 | 95.36 | 85.72 | 87.48 |
| Noise-Vector | 51.80 | 23.04 | 51.28 | 26.80 | 54.92 | 8.64 | 52.44 | 17.88 |
| Label-Anchor | 86.00 | 38.96 | 80.36 | 83.52 | 58.72 | 73.68 | 71.32 | 56.44 |
| Task-Vector | 84.92 | 36.48 | 71.92 | 73.52 | 65.00 | 76.48 | 84.80 | 56.04 |
| ICV | 90.16 | 35.36 | 55.52 | 73.28 | 73.68 | 90.08 | 85.12 | 59.08 |
| I²CL | 90.28 | 43.76 | 75.44 | 84.24 | 76.40 | 91.87 | 85.52 | 76.36 |
| AutoComp | 91.11 | 27.35 | 59.15 | 85.74 | 58.84 | 91.18 | 84.33 | 61.51 |
| ICAE | 89.79 | 40.25 | 51.04 | 80.47 | 67.73 | 59.93 | 86.77 | 50.44 |
| CEPE | 75.83 | 35.35 | 55.36 | 79.44 | 60.83 | 85.56 | 82.75 | 59.34 |
| I³CL (Ours) | **92.66** | **51.16** | **87.36** | **87.07** | **86.28** | **95.88** | **90.67** | **90.84** |

Table 1: Accuracy comparison between our approach and baseline methods on LLaMA2-7B across eight classification tasks under the 5-shot few-shot setting. The best results are highlighted in bold, and the second-best results are underlined.

## 4.2 MAIN RESULT

The performance comparison across different few-shot classification tasks is summarized in Table 1. Both few-shot ICL and LoRA finetuning substantially outperform zero-shot learning, confirming

| Method | Toxicity (%) | ROUGE-1 | BERT Score |
|---|---|---|---|
| Original Test Set | 84.58 | - | - |
| Gold-standard Reference | 16.23 | - | - |
| Zero-Shot | 71.6 | 73.15 | 93.32 |
| ICL | 66.81 | 74.19 | 93.11 |
| LoRA FT | 48.94 | 57.32 | 89.34 |
| ICV | 39.54 | 65.97 | 92.73 |
| $I^3$CL | 57.56 | 70.55 | 90.79 |
| $I^3$CL (Ours) | **24.80** | 69.19 | 91.74 |

Table 2: Performance comparison between our method and baselines on ParaDetox (LLaMA2-7B and 5-shot demonstrations).

that even a small set of demonstrations can markedly improve LLM performance. LoRA finetuning generally achieves the best results, but this comes at the expense of large computational cost due to parameter updates.

Among implicit in-context learning methods, $I^2$CL achieves state-of-the-art performance across these datasets. However, its reliance on independent demonstration encoding and average pooling limits its ability to capture inter-demonstration interactions and adaptive importance weighting—core mechanisms driving standard ICL's success. This limitation is particularly evident in $I^2$CL's underperformance compared to standard ICL on datasets such as SST-2, SST-5 and MR.

Our proposed method, $I^3$CL, directly addresses these limitations through two lightweight interaction modules. These modules reintroduce the critical capabilities of standard ICL within the implicit learning framework. Results demonstrate that $I^3$CL outperforms all existing implicit ICL methods across all eight evaluated datasets. Notably, $I^3$CL even surpasses performance of LoRA fine-tuning (LoRA-FT) methods on several tasks. When compared against recent prompt compression methods, $I^3$CL also delivers consistent improvements. These results highlight the superiority of $I^3$CL. Full experimental results on other backbones are provided in Appendix A.5.

Regarding style transfer performance (Table 2), $I^3$CL achieves significantly lower toxicity (24.80%) than all baselines on the ParaDetox dataset, substantially outperforming even current SOTA ICV (39.54%). $I^3$CL also maintains strong performance in fluency (ROUGE-1) and semantic preservation (BERT Score), matching or exceeding previous baselines. Besides, $I^3$CL's improvement over $I^2$CL highlights that the latter's static, coarse-grained context aggregation is insufficient for this generative task. This insufficiency is further evidenced by $I^2$CL's high average training loss (2.70). In contrast, $I^3$CL's dynamic, interaction-aware mechanism leads to lower average loss of 0.67, enabling superior performance on this generative task. These results demonstrate $I^3$CL's effectiveness on more complex, non-classification tasks.

| Method / Dataset (Avg Len) | SST-2 246.01 | SST-5 370.31 | TREC 653.63 | AGNews 343.96 | HS18 512.29 | DBPedia 475.10 | MR 366.94 | Subj 637.94 |
|---|---|---|---|---|---|---|---|---|
| $I^2$CL | 90.28 | 43.76 | 75.44 | 84.24 | 76.40 | 91.87 | 85.52 | 76.36 |
| + Multi-Token | 91.16 | 44.08 | 75.64 | 84.64 | 85.36 | 92.08 | 86.44 | 78.60 |
| + Demo-Interaction | 92.56 | 44.56 | 77.16 | 84.76 | 82.96 | 94.48 | 86.64 | 77.52 |
| + Demo-Importance | 92.32 | 45.32 | 86.56 | 86.44 | 82.28 | 93.52 | 87.00 | 87.20 |
| + All ($I^3$CL) | **92.66** | **51.16** | **87.36** | **87.07** | **86.28** | **95.88** | **90.67** | **90.84** |
| $I^3$CL w/o FC | 91.40 | 44.20 | 77.60 | 85.27 | 85.15 | 92.67 | 86.27 | 79.96 |

Table 3: Accuracy performance across different ablation variants of $I^3$CL, illustrating the impact of each component.

## 4.3 ABLATION STUDY

To assess the contribution of each component, we extend $I^2$CL with three targeted enhancements: multi-token representation ("+ Multi-Token"), inter-demonstration interaction ("+ Demo-

Interaction"), and demonstration-importance estimation ("+ Demo-Importance"). As shown in Table 3, all components individually outperform the $I^2CL$ baseline. Notably, multi-token representations yield consistent gains across all datasets, with substantial improvements on tasks involving longer sequences (e.g., TREC: +4.20, HS18: +6.96, Subj: +4.24), demonstrating their effectiveness in addressing the limited context captured by a single latent token. Among all modules, "+ Demo-Importance" brings the largest overall improvement, while combining it with "+ Demo-Interaction" forms the complete $I^3CL$ framework, which achieves the best performance. Even when removing the additional fully connected layers used for the $q$, $k$, and $v$ projections ("$I^3CL$ w/o FC"), the model still surpasses the baseline, confirming that the structural design of $I^3CL$ is intrinsically strong, with the FC layers further enhancing generalization.

## 4.4 FURTHER ANALYSIS

**Impact of Different Strategies on Target Module**. We further explore alternative strategies for both the token selection and inter-demonstration interaction modeling. As illustrating as Figure 3, for the token selection, we experimented with the following strategies: (1) keeping only the last token from both the prompt and label segments ("Last-Token"), (2) randomly selecting a token from each segment ("Random-Token"), (3) selecting the token with the highest weight from each segment (Max-Pooling), and (4) aver-

| Target Module: Token Selection Strategy for Sentence | | | |
|---|---|---|---|
| Strategy | SST-5 | TREC | Subj |
| Last-Token | **51.16** | **87.36** | **90.84** |
| Random-Token | 48.88 | 85.40 | 88.80 |
| Max-Pooling | 24.56 | 25.88 | 51.32 |
| Avg-Pooling | 44.08 | 85.85 | 80.07 |

| Target Module: Demonstration Interaction Strategy | | | |
|---|---|---|---|
| Strategy | SST-5 | TREC | Subj |
| Standard Attention | 51.16 | 87.36 | 90.84 |
| Multi-Head Attention | **52.73** | **87.93** | **92.53** |

Figure 3: Impact of different strategies on the target modules.

aging all tokens into a single representation (Avg-Pooling). The results show that the "Last-Token" strategy performs best, significantly outperforming "Random-Token", "Max-Pooling", and "Avg-Pooling". This suggests that the final token position in both segments carries more discriminative task-related information in the context. In contrast, the Max-Pooling strategy performs poorly, possibly because the token with the highest weight does not necessarily contain sufficient semantic information and may instead act as a high-impact noise token. For the interaction modeling strategy, Multi-Head Attention clearly outperforms the standard attention mechanism, achieving consistent improvements on the three tasks. This demonstrates the superior expressive power of the multi-head mechanism in capturing relationships among demonstrations. Together, these results validate the effectiveness and soundness of the design choices made in both modules.

| Set A \ Set B | SST-2 | SST-5 | TREC | AGNews | HS18 | DBPedia | MR | Subj |
|---|---|---|---|---|---|---|---|---|
| SST-2 | **98.91** | 97.54 | 85.21 | 86.42 | 87.47 | 85.44 | 98.49 | 91.27 |
| SST-5 | 96.00 | **99.84** | 85.36 | 88.04 | 88.86 | 87.41 | 98.30 | 92.07 |
| TREC | 85.92 | 85.42 | **99.91** | 86.69 | 79.89 | 88.07 | 83.50 | 83.55 |
| AGNews | 84.78 | 87.05 | 85.51 | **99.47** | 82.27 | 86.83 | 86.36 | 87.00 |
| HS18 | 85.85 | 88.62 | 80.09 | 82.41 | **99.52** | 83.59 | 87.87 | 91.37 |
| DBPedia | 85.35 | 87.56 | 87.95 | 87.45 | 83.81 | **99.67** | 85.59 | 85.71 |
| MR | 97.19 | 98.39 | 84.24 | 86.88 | 88.25 | 85.67 | **99.60** | 91.85 |
| Subj | 89.65 | 91.48 | 83.72 | 87.78 | 90.60 | 85.49 | 91.63 | **99.35** |

Table 4: $I^2CL$ Task vector similarity across different demonstration sets.

**Efficiency Analysis**. We evaluate our method and several baselines under the same setting using the TREC dataset and LLaMA2-7B, tracking calibration time, inference overhead, and peak memory usage. As shown in Figure 4, We make three key observations: **(1) Calibration efficiency**. In standard few-shot settings (5-30 shots), $I^3CL$ increases ≤7% overhead compared to $I^2CL$, demonstrating comparable efficiency. While 100-shot calibration increases by 34% (reflecting interactive complexity), this one-time offline cost is justified by significant accuracy gains (+11.92%). **(2) Inference Speed**. In critical few-shot scenarios (5-30 shots) show ≤1.3% increase in inference latency. Al-

| Set A\Set B | SST-2 | SST-5 | TREC | AGNews | HS18 | DBPedia | MR | Subj |
|---|---|---|---|---|---|---|---|---|
| SST-2 | **99.99** | 65.40 | -0.50 | 65.78 | 67.51 | 61.86 | 97.88 | 71.74 |
| SST-5 | 67.30 | **95.43** | 16.36 | 57.74 | 59.25 | 55.41 | 68.15 | 58.52 |
| TREC | -0.58 | 17.59 | **99.92** | 0.52 | 0.43 | 0.89 | -0.57 | 0.14 |
| AGNews | 65.59 | 56.40 | 0.50 | **98.41** | 63.96 | 66.90 | 66.97 | 67.21 |
| HS18 | 67.37 | 57.45 | 0.46 | 63.89 | **97.33** | 56.63 | 68.41 | 70.67 |
| DBPedia | 62.23 | 55.04 | 0.89 | 66.96 | 56.95 | **99.36** | 63.16 | 60.09 |
| MR | 97.88 | 66.24 | -0.50 | 67.07 | 68.54 | 62.75 | **98.99** | 72.97 |
| Subj | 71.77 | 57.47 | 0.20 | 67.06 | 70.97 | 59.94 | 73.03 | **97.05** |

Table 5: I$^3$CL Task vector similarity across different demonstration sets.

though overhead rises at 100-shot (4.91x vs. I$^2$CL's 4.08x), it remains dramatically lower than standard ICL. **(3) Memory Efficiency.** I$^3$CL's peak memory usage closely matches I$^2$CL's. Crucially, I$^3$CL avoids Out-Of-Memory (OOM) errors in all tested scenarios, unlike standard ICL which fails at 100-shot. This demonstrates I$^3$CL's superior scalability.

**I$^2$CL vs. I$^3$CL Task Vectors**. We sampled distinct demonstration sets from two separate datasets (A and B). We then computed the cosine similarity for all possible task combinations. Table 4 and Table 5 represent the results for I$^2$CL and I$^3$CL, respectively; bold diagonal values indicate intra-task similarity. We observe that I$^2$CL produces indistinct task representations, evidenced by its high inter-task similarity (typically exceeding 85%) alongside its high intra-task consistency (>98% diagonal). This indicates significant contamination by task-irrelevant features, hindering clear task separation. This is notably observed in the high SST-2/TREC similarity (85.21%). In contrast, I$^3$CL captures distinct task-specific patterns, maintaining strong

| Method | Acc | 5-Shot | 10-Shot | 30-Shot | 100-Shot |
|---|---|---|---|---|---|
| Calibration Time Comparison | | | | | |
| I$^2$CL | 75.44 | 1x | 1x | 1x | 1x |
| I$^3$CL | 87.36 | 1.02x | 1.03x | 1.07x | 1.34x |
| Inference Time Comparison | | | | | |
| Zero-Shot | 41.96 | 1x | 1x | 1x | 1x |
| ICL | 60.08 | 5.84x | 9.93x | 36.66x | – (OOM) |
| I$^2$CL | 75.44 | 4.08x | 4.08x | 4.08x | 4.08x |
| I$^3$CL | 87.36 | 4.10x | 4.13x | 4.27x | 4.91x |
| Peak Memory Usage | | | | | |
| Zero-Shot | 41.96 | 1x | 1x | 1x | 1x |
| ICL | 60.08 | 1.59x | 2.15x | 5.84x | – (OOM) |
| I$^2$CL | 75.44 | 2.04x | 2.04x | 2.04x | 2.04x |
| I$^3$CL (Ours) | 87.36 | 2.15x | 2.25x | 2.45x | 2.83x |

Figure 4: Calibration time, Inference Time and Peak memory usage comparison under different numbers of shots.

intra-task consistency (>95% diagonal) while achieving sharp inter-task discrimination. For unrelated tasks (e.g., SST-2 vs. TREC), similarity drops to near-zero levels. For related tasks (e.g., SST-2 vs. SST-5) with differing classification goal, similarity remains moderated (65.40%), significantly lower than I$^2$CL's 97.54%. This underscores I$^3$CL's ability to encode tasks with greater nuance.

## 5 CONCLUSION

This work identifies key limitations in implicit in-context learning (I$^2$CL): while prioritizing inference speed, it neglects crucial mechanisms underpinning standard ICL's effectiveness—inter-demonstration and demonstration-query interactions. To address this, we introduce Interactive Implicit In-Context Learning (I$^3$CL), a simple yet powerful framework that incorporates two lightweight interaction modules to restore standard ICL's core abilities while preserving the efficiency of implicit ICL methods. Extensive experiments on four LLMs and nine diverse tasks demonstrate that I$^3$CL consistently outperforms prior implicit ICL approaches, achieving significant accuracy improvements with efficiency comparable to I$^2$CL. Ablation studies confirm the substantial contribution of each component. Furthermore, detailed analysis shows that I$^3$CL produces a more expressive and discriminative task vector than I$^2$CL. Motivated by recent findings on the complementary roles of similarity and diversity in demonstration selection (Mavromatis et al., 2023), we plan to incorporate diversity-aware strategies into the demonstration–query interaction, as a promising direction for future improvement.

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

# A   APPENDIX

## A.1   IMPLEMENTATION DETAILS

For few-shot setting, we follow the I$^2$CL (Li et al., 2025), the current SOTA approach in this area, by randomly sampling five demonstrations per class to mitigate majority-label bias and achieve more robust few-shot performance. Moreover, we employ the HuggingFace version of the data (Lhoest et al., 2021) for evaluation. The task instructions and manually designed prompt templates are provided in Figure 8. During evaluation, we report the average accuracy across eight tasks under five different random seeds. For the calibration process, we optimize the coefficients for 100 epochs using the AdamW optimizer (Loshchilov & Hutter, 2017) on the same set of demonstrations. The hyperparameter setting are listed in Table 6.

| Model | Batch Size | Learning Rate |
|---|---|---|
| LLaMA2-7B | 2 | $2e^{-4}$ |
| LLaMA3-8B | 2 | $2e^{-5}$ |
| GLM4-9B | 2 | $2e^{-6}$ |
| DeepSeek-R1 | 2 | $2e^{-4}$ |

Table 6: The hyperparameter configurations for LLMs. "Batch Size" refers to the training batch size, while "Learning Rate" denotes the initial learning rates for LLMs, which is decayed gradually during training.

## A.2   ABLATION STUDY ON THE NECESSITY OF TASK INSTRUCTION FOR EACH DEMONSTRATION

| Method | SST-2 | SST-5 | TREC | AGNews | HS18 | DBPedia | MR | Subj | Avg. |
|---|---|---|---|---|---|---|---|---|---|
| I$^3$CL | 92.66 | 51.16 | 87.36 | 87.07 | 86.28 | 95.88 | 90.67 | 90.84 | 85.24 |
| w/o DI | 91.20 | 49.00 | 86.40 | 85.68 | 84.20 | 93.40 | 87.82 | 82.68 | 82.55 |

Table 7: Ablation study on the effect of encoding task instructions for each demonstration. "w/o DI" indicates that task instructions are removed from the demonstrations.

To preserve the semantic completeness of each demonstration, I$^3$CL encodes each example in isolation, requiring the task instruction $I$ to supply essential contextual grounding. Without $I$, the LLM often struggles to relate the demonstration's content to the intended task objective. As shown in Table 7, removing per-demonstration instructions (w/o DI) consistently leads to performance degradation, with the average accuracy dropping from 85.24% to 82.55%. This confirms that the instruction $I$ plays a crucial role in enabling the model to extract task-relevant and high-quality representations from demonstrations.

## A.3   ROBUSTNESS VISUALIZATION ANALYSIS

Figure 5 compares the representations of I$^2$CL and our proposed I$^3$CL on the Subj and TREC tasks under 0% and 50% noise. Here, "noise" refers to standard Gaussian noise vectors injected into the context vectors at varying magnitudes to simulate irrelevant information in the demonstration set relative to the current query, allowing us to evaluate the robustness of the methods. I$^2$CL produces entangled clusters and degrades significantly under noisy conditions, whereas I$^3$CL consistently yields more compact and well-separated clusters, maintaining clear decision boundaries even with high noise. This demonstrates that I$^3$CL achieves better generalization and robustness, as it leverages attention mechanisms to adaptively select the most suitable demonstration vectors for prediction.

## A.4   CASE STUDY

To illustrate how the softmax-based attention mechanism in I$^3$CL guides the selection of relevant demonstrations for a given test sample, we conducted a focused case study on six test questions from

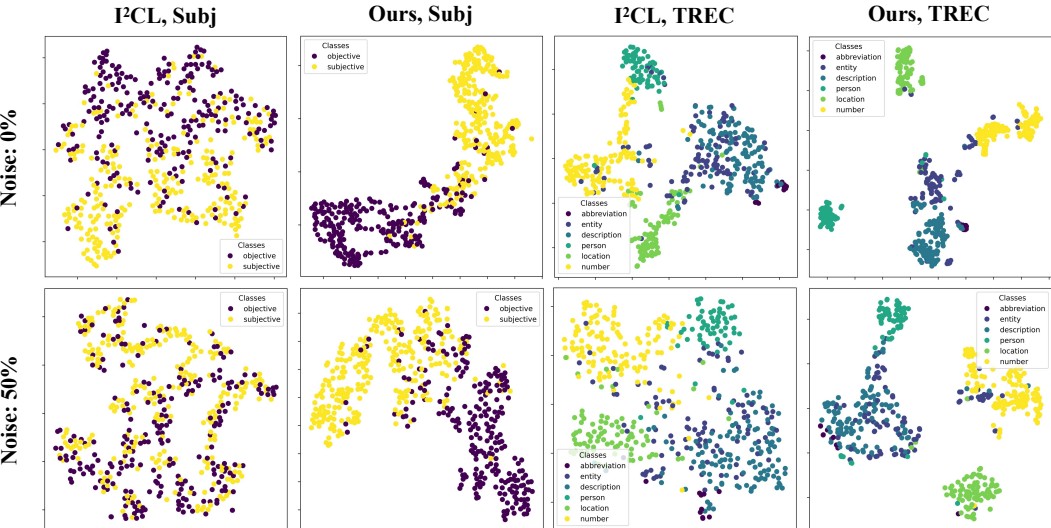

Figure 5: t-SNE visualizations on the Subj and TREC tasks. We compare I²CL with our method under different noise levels.

| Test Samples | Person | Number | Description | Location | Abbreviation | Entity |
|---|---|---|---|---|---|---|
| Q: What state has the least amount of rain per year? (Location) | 0.136 | **0.170** | 0.165 | 0.168 | 0.154 | 0.167 |
| Q: Who was the first African American to play for the Brooklyn Dodgers? (Person) | **0.181** | 0.166 | 0.162 | 0.164 | 0.171 | 0.165 |
| Q: What is pilates? (Description) | 0.166 | 0.159 | **0.184** | 0.160 | 0.168 | 0.182 |
| Q: When was the Boston tea party? (Number) | 0.139 | **0.188** | 0.169 | 0.182 | 0.155 | 0.167 |
| Q: What is a mirror made out of? (Entity) | 0.165 | 0.167 | 0.167 | 0.166 | 0.165 | **0.169** |
| Q: What is TUJ? (Abbreviation) | 0.169 | 0.166 | 0.165 | 0.162 | **0.172** | 0.166 |

Table 8: Case Study: Attention Weights (Post-Softmax) for Demonstrations Across Test Samples

the TREC dataset. The demonstration set was carefully constructed to cover six distinct question categories, including Person, Number, Description, Location, Abbreviation, and Entity, with one representative example per category:

- **Person**: What famous husband-and-wife team did radio ads for Blue Nun wine? → Label: Person.

- **Number**: What's men's par on a 455-yard golf hole? → Label: Number

- **Description**: What function does a community's water tower serve? → Label: Description

- **Location**: Where is Belize located? → Label: Location

- **Abbreviation**: What does R.E.M. stand for, as in the rock group R.E.M.? → Label: Abbreviation

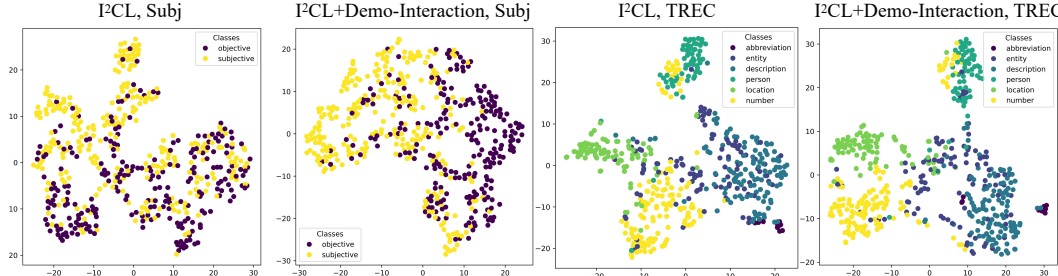

Figure 6: t-SNE visualization of learned embeddings from I$^2$CL and I$^2$CL with "Demo-interaction" module across the Subj and TREC datasets.

- **Entity**: What was the name of Roy Rogers's dog? → Label: Entity

For each test question, we visualized the post-softmax attention weights that I$^3$CL assigned to all six demonstrations. As shown in Table 8, the results demonstrate that I$^3$CL dynamically adjusts attention weights based on the input question. In each case, the demonstration most semantically aligned with the test question receives the highest weight. For example, the "Location" demonstration has the highest weight for the question about the state with the least rainfall, while the "Person" demonstration is most emphasized for the question about the first African American baseball player. This case study highlights the effectiveness of I$^3$CL's importance-weighting mechanism, showing that the model can selectively prioritize the most relevant demonstrations from a small, diverse set, thereby improving its ability to classify questions accurately.

| Method | SST-2 | SST-5 | TREC | AGNews | HS18 | DBPedia | MR | Subj |
|---|---|---|---|---|---|---|---|---|
| Zero-Shot | 88.40 | 47.84 | 67.84 | 70.60 | 63.00 | 78.60 | 82.62 | 71.96 |
| ICL | 92.49 | 50.76 | 72.24 | 74.80 | 52.86 | 87.67 | _87.27_ | 74.53 |
| LoRA-FT | **94.26** | _52.48_ | _85.64_ | **87.63** | 86.01 | **96.83** | 86.21 | _91.22_ |
| Noise-Vector | 51.83 | 29.23 | 24.47 | 27.28 | 57.81 | 7.87 | 52.20 | 47.42 |
| Label-Anchor | 92.45 | 45.86 | 34.62 | 69.84 | 73.93 | 84.28 | 73.75 | 54.83 |
| Task-Vector | 91.85 | 45.65 | 54.27 | 85.55 | 80.60 | 67.66 | 82.47 | 53.82 |
| ICV | 88.70 | 40.68 | 58.62 | 84.58 | 90.15 | 92.83 | 81.27 | 51.43 |
| I$^2$CL | 89.63 | 48.64 | 76.00 | 85.83 | _90.60_ | 93.47 | 82.85 | 77.12 |
| I$^3$CL (Ours) | _93.43_ | **53.60** | **88.28** | _85.89_ | **91.60** | _94.16_ | **88.20** | **92.08** |

Table 9: Performance Comparison on LLaMA3-8B

| Method | SST-2 | SST-5 | TREC | AGNews | HS18 | DBPedia | MR | Subj |
|---|---|---|---|---|---|---|---|---|
| Zero-Shot | 94.20 | 50.84 | 69.60 | 86.21 | 83.43 | 88.83 | 89.28 | 54.04 |
| ICL | 95.45 | 51.64 | 77.88 | 86.67 | 84.21 | 97.40 | 89.41 | 61.16 |
| LoRA-FT | _95.61_ | _51.65_ | **93.09** | **91.19** | _85.18_ | 97.85 | **91.40** | _82.21_ |
| Noise-Vector | 48.27 | 23.64 | 18.88 | 25.81 | 47.82 | 7.81 | 54.62 | 55.18 |
| Label-Anchor | 90.88 | 49.46 | 53.43 | 48.41 | 71.64 | 83.29 | 81.04 | 53.80 |
| Task-Vector | 95.85 | 48.42 | 51.27 | 68.64 | 74.41 | 70.87 | 82.25 | 62.80 |
| ICV | 94.36 | 49.68 | 69.20 | 84.46 | 80.81 | 96.21 | 86.83 | 63.27 |
| I$^2$CL | 94.63 | 51.00 | 77.84 | 88.60 | 81.49 | 97.86 | 88.10 | 59.60 |
| I$^3$CL (Ours) | **95.92** | **52.36** | _92.76_ | _89.66_ | **87.81** | **98.24** | _90.04_ | **82.40** |

Table 10: Performance Comparison on GLM4-9B

## A.5 PERFORMANCE COMPARISON ON OTHER LLMS

We provide comprehensive experimental results on three widely used model backbones—LLaMA3-8B (Dubey et al., 2024), GLM4-9B (Zeng et al., 2024), and DeepSeek-R1-8B (Guo et al., 2025).

| Method | SST-2 | SST-5 | TREC | AGNews | HS18 | DBPedia | MR | Subj |
|---|---|---|---|---|---|---|---|---|
| Zero-Shot | 86.63 | 30.80 | 45.40 | 80.21 | 80.27 | 87.31 | 73.81 | 53.64 |
| ICL | 91.10 | 39.28 | 68.40 | 81.22 | 82.26 | 90.80 | 85.25 | 55.56 |
| LoRA-FT | 91.86 | 47.51 | **90.39** | 88.20 | 89.30 | **97.82** | **86.85** | 64.83 |
| Noise-Vector | 51.80 | 19.83 | 19.41 | 26.86 | 51.23 | 8.28 | 52.37 | 52.65 |
| Label-Anchor | 86.46 | 45.84 | 54.29 | 66.49 | 56.48 | 75.65 | 85.87 | 50.51 |
| Task-Vector | 86.53 | 29.86 | 66.80 | 83.45 | 83.68 | 64.97 | 83.46 | 47.35 |
| ICV | 87.88 | 28.02 | 41.87 | 80.87 | 85.04 | 81.87 | 81.82 | 63.27 |
| I$^2$CL | 81.84 | 33.00 | 73.26 | 84.63 | 86.26 | 88.64 | 80.82 | 61.64 |
| I$^3$CL (Ours) | **91.91** | **49.55** | 89.60 | **89.69** | **90.74** | 96.45 | 86.46 | **76.00** |

Table 11: Performance Comparison on DeepSeek-R1-8B

As reported in Tables 9, 10, and 11, all results are averaged over five independent runs to ensure reliability.

Across all backbones and tasks, our method (I$^3$CL) consistently achieves the best performance, substantially outperforming strong baselines such as Zero-Shot, ICL, and I$^2$CL, demonstrating robust generalization and strong cross-model transferability. For example, on LLaMA3-8B, I$^3$CL surpasses I$^2$CL by 4.96%, 12.28%, and 14.96% on SST-5, TREC, and Subj, respectively, with similar improvements observed on GLM4-9B and DeepSeek-R1. These results confirm the effectiveness of our approach in leveraging demonstration information across diverse model architectures.

## A.6 SCALABILITY OF I$^3$CL

As shown in Figure 7, although I$^3$CL is designed for few-shot settings, its performance continues to improve with more demonstrations and peaks around 30 shots. This indicates that I$^3$CL not only excels in low-data regimes but also scales effectively, demonstrating strong robustness. In contrast, I$^2$CL shows only minor gains as demonstrations increase due to its inability to model inter-example interactions or adaptively weight inputs, limiting its use of additional information. Standard ICL performs even worse under this task's semantic complexity and often degrades with too many demonstrations, revealing both inefficiency and sensitivity to context length. Overall, the results highlight that I$^3$CL achieves a favorable balance between efficiency and scalability: its lightweight attention preserves near-

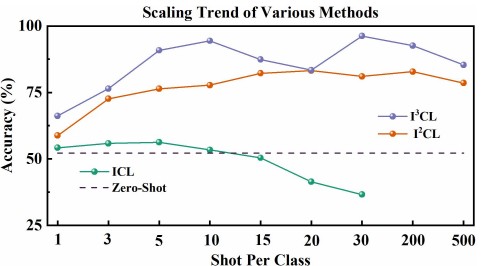

Figure 7: Comparison of the scaling ability of different methods, all evaluated on the Subj dataset using the LLaMA2-7B model.

standard inference speed while restoring the analytical and selective capabilities of ICL, enabling consistent benefits from more demonstrations—something neither I$^2$CL nor standard ICL can offer.

## A.7 FURTHER VISUALIZATION ANALYSIS FOR "DEMO-INTERACTION" MODULE

To visualize this expressive capability, we generated t-SNE plots of the learned embeddings, as shown in Figure 6. The results show that while the I$^2$CL model produces representations with significant class overlap and indistinct boundaries, incorporating the Demo-Interaction module yields clearer class separation. This visualization evidence confirms that enhanced interactions produce more discriminative and expressive representations.

## A.8 REPRODUCTION OF BASELINE METHODS

Since the test set used in I$^2$CL (Li et al., 2025) is randomly constructed by sampling 500 instances from the original test set for each task, and neither the specific sampled data nor the random seed has been released. To ensure a fair comparison, we adopt the same sampling strategy as I$^2$CL to resample the test set for each task. Based on resampled test sets, we reproduce all baseline methods.

| Task | Prompt Template |
|------|-----------------|
| SST-2, SST-5, MR | <Start of Instruction> You are given a list of sentiment labels: {label_list}.\nYour task is to analyze the given sentence and determine the most appropriate sentiment label from the list based on its context. <End of Instruction> \n\n <Query> Given Sentence: {text}\nPlease determine which sentiment of {label_list} based on the context. \n ASSISTANT: The sentiment of \'{text}\' is |
| AGNews | <Start of Instruction> You are a news classifier. Given a news, classify it into one of the following types: {label_list}. <End of Instruction> \n\n <Query> Given News: {text}\nPlease determine which type of {label_list} for the given news based on the context. \n ASSISTANT: The type of \'{text}\' is |
| DBPedia | <Start of Instruction> You are an AI language model trained to determine the semantic type of the entity mentioned in the sentence. Classify the entity into one of 14 types: {label_list} <End of Instruction> \n\n <Query> Given Sentence: {text}\n Entity: {aspect}.\nPlease determine which type of {label_list} for the entity based on the context. \n ASSISTANT: The type of \'{aspect}\' is |
| HateSpeech | <Start of Instruction> You are an AI language model trained to detect hate speech language. Classify the following sentence into one of two categories: {label_list}. <End of Instruction> \n\n <Query> Given Sentence: {text}\nPlease determine which category of {label_list} based on the context. \n ASSISTANT: The category of \'{text} is |
| Subj | <Start of Instruction> You are an AI model trained to complete the subjectivity classification. Classify the following sentence into one of two subjectivity types: {label_list. <End of Instruction> \n\n <Query> Given Sentence: {text}\nPlease determine which subjectivity type of {label_list} for the given sentence based on the context. \n ASSISTANT: The subjectivity type of \'{text}\' is |
| TREC | <Start of Instruction> You are a question classifier. Given a question, classify it into one of the following types: {label_list}. <End of Instruction> \n\n <Query> Given Question: {text}\nPlease determine which type of {label_list} for the given question based on the context. \n ASSISTANT: The type of the question is |
|  |  |

Figure 8: Prompt templates specific to each task are employed consistently across all experiments.

The detailed reproduction process is described below. To support reproducibility and facilitate future research, the sampled test data will be made available in our source code.

**Zero-Shot** (Li et al., 2025): In this baseline, we directly perform inference on the test samples using the LLM.

**ICL** (Li et al., 2025): For each class, we randomly sample five demonstrations from the training set and incorporate them into the prompt to condition the LLM for inference on the test samples.

**LoRA-FT** (Shen et al., 2022): Similar to the ICL, this baseline fine-tunes the LLM using LoRA on the same set of demonstrations. The fine-tuning parameters are listed in Table **??**.

**Noise-Vector** (Li et al., 2025): In this baseline method, we follow $I^2CL$ (Li et al., 2025) to replace context vectors with random noises while keeping all other settings identical to $I^2CL$.

**Label-Anchor** (Wang et al., 2023): We directly test the officially released model from here: https://github.com/lancopku/label-words-are-anchors on our tasks. All hyper-parameters are set by default values. Evaluation protocols remain the same as ours.

**Task-Vector** (Hendel, 2023): We replicate the task vector method via the code released by $I^2CL$ from here: https://github.com/LzVv123456/I2CL. All hyper-parameters are set by default values. Evaluation protocols remain the same as ours.

**ICV** (Liu et al., 2024): We replicate the ICV method via the code released by $I^2CL$ from here: https://github.com/LzVv123456/I2CL. All hyper-parameters are set by default values. Evaluation protocols remain the same as ours.

**$I^2CL$** (Li et al., 2025): We directly test the officially released model from here: https://github.com/LzVv123456/I2CL on our tasks. All hyper-parameters are set by default values. Evaluation protocols remain the same as ours.

**AutoComp** (Chevalier et al., 2023): We directly test the officially released model from here: https://github.com/princeton-nlp/AutoCompressors on our tasks. All hyper-parameters are set by default values. Evaluation protocols remain the same as ours.

**ICAE** (Ge et al., 2023): We directly test the officially released model from here: https://github.com/getao/icae on our tasks. All hyper-parameters are set by default values. Evaluation protocols remain the same as ours.

**CEPE**(Yen et al., 2024): We directly test the officially released model from here: https://github.com/princeton-nlp/CEPE on our tasks. All hyper parameters are set by default values. Evaluation protocols remain the same as ours.

