# OpenReview forum: "Interactive Implicit In-context Learning"
_ICLR.cc/2026/Conference — Submitted to ICLR 2026_

### Official Review · Reviewer_22Y8 · 2025-10-22

**Soundness:** 1
**Presentation:** 2
**Contribution:** 1
**Rating:** 2
**Confidence:** 5

**Summary:**

This paper proposes $I^3CL$ based on the previous I2CL framework, with some module modification focusing on the anisotropy of demonstration representations (i.e., context vector). Experiment results show that $I^3CL$ is a new SotA among the hidden state steering methods on zero-shot inputs.

**Strengths:**

1. The two issues of the prior work (I2CL), i.e., no demonstration interaction and the naive averaging problem, pointed out by this paper, are reasonable. Therefore, the paper proposes $I^3CL$ to intuitively address these limitations.

2. The experiments show that the proposed $I^3CL$ method achieves SotA performance among implicit ICL methods. Moreover, the ablation analysis confirms the effectiveness of its components.

**Weaknesses:**

1. Many claims or assumptions remain unverified, and some heuristic designs lack sufficient motivation.

    1. Line 084: “exchange, contrast, and complement information to form a more expressive collective representation” is not directly supported by experimental evidence. I expect a direct measurement of such expressive ability.

    2. Line 195: “we compute a representation by averaging two specific last-token representations: the last token of the context (prompt) part and the last token of the label part” is a heuristic design. I can not get the motivation for extracting the tokens from these two positions and averaging them, nor convincing experimental results to justify the necessity of doing so. Specifically, one can expect that the last token of the label already encodes the preceding demonstration information, so extracting only the label token’s hidden states and MLP outputs might suffice. You might conduct more ablation experiments to verify whether extracting both the label token and the demonstration token is truly necessary.

    3. Line 089: “enabling the model to selectively emphasize the most informative demonstrations” is also not directly supported by experimental evidence. At least, I would like to see a case study of the selected demonstrations by the softmax processing. Also, another concern is raised on this point, see below.

2. The demonstrations are weighted using softmax and cosine similarity.

    1. This essentially reproduces a demonstration selection process, which can benefit not only from the similarity but also from the diversity of demonstrations [1]. Since evaluating diversity does not significantly affect computational efficiency, I suggest that the authors incorporate diversity into their demonstration-to-query interaction process. Furthermore, the authors could even consider parameterizing these softmax processes, since gradient-based optimization is already employed for efficient calibration, adding a few more parameters would be acceptable.

    2. It is conceivable that, in classification tasks, a demonstration similar to the query is likely to have the same label as the query. Directly copying the hidden states of this consistent label into the zero-shot inference naturally biases the output toward that label, which may cause your method to degenerate into a KNN retriever operating on embedded representations. An intuitive way to address this concern is to conduct experiments on non-classification tasks, as described below.

3. The experiments are insufficient. The proposed method is clearly not limited to classification tasks, yet this paper only evaluated it on classification datasets. Moreover, although four models are listed in the paper, this paper conducted a complete evaluation on only one of them. This undermines the credibility of their approach.

4. This work is incremental. Although I believe the improvements made in this paper are reasonable, it still largely follows the framework of Li et al. Personally, I do not object to incremental work, but I am unsure whether it fits the standards of ICLR. I therefore raise this point for the AC’s consideration, though it does not affect my main score.

5. Some writing issues, see Question.


[1] Which Examples to Annotate for In-Context Learning? Towards Effective and Efficient Selection. [https://arxiv.org/abs/2310.20046v1](https://arxiv.org/abs/2310.20046v1)

**Questions:**

1. Table 5 (Fig. 5 in your notation): As far as I understand, compared with $I^3CL$, each step in I2CL uses a simpler module variant (e.g., average pooling instead of self-attention). Why, then, is its time cost greater than that of $I^3CL$?

2. Line 218: I did not fully understand what “global collective regularities” specifically refers to. I can imagine that you might be referring to something similar to a “task representation,” but I would like to know what these “regularities” actually are in numerical terms. Therefore, you may need to provide a more detailed analysis comparing the task vector extracted by I2CL with that of $I^3CL$, highlighting their semantic differences.

3. Line 414: What does the “highest weight” refer to? Is the weight the norm of the hidden state?

4. Line 456: What does “noise” refer to here?

5. Line 617: Typo.

---

> ### Author Response · Authors · 2025-11-29
> **The first reply to Reviewer 22Y8**
>
> We sincerely thank you for taking the time to review our work!
>
> **1.Line 084: “exchange, contrast, and complement information to form a more expressive collective representation” is not directly supported by experimental evidence. I expect a direct measurement of such expressive ability. (from weakness 1.1)**
>
> Thank you for your valuable suggestion regarding the experimental evidence.
>
> To address this point, we performed an ablation study evaluating the isolated impact of the Demo-Interaction module. The results are presented in the table below:
>
> | Method | SST-2 | SST-5 | TREC | AGNews | HS18 | DBPedia | MR | Subj |
> |--------|-------|-------|------|--------|------|---------|-----|------|
> | I2CL   |90.28  |43.76  |75.44 |84.24   |76.40 |91.87    |85.52|76.36 |
> | + Demo-Interaction |92.56|44.56|77.16|84.76|82.96|94.48|86.64|77.52|
>
> Adding this module consistently outperforms the I2CL baseline across all 8 downstream tasks. **The observed accuracy gains (e.g., +2.28 on SST-2, +6.56 on HS18,and +2.61 on DBPedia) demonstrate that the module learns more expressive collective representations.**
>
> To visualize this expressive capability, we generated t-SNE plots of the learned embeddings. The results show that **while the I2CL model produces representations with significant class overlap and indistinct boundaries, incorporating the Demo-Interaction module yields clearer class separation.** This visualization evidence confirms that enhanced interactions produce more discriminative and expressive representations.
>
> Due to space constraints within the rebuttal, we are unable to include the t-SNE figures. The ablation results and full set of t-SNE visualizations will be included in the revised manuscript.
>
> (*Due to the word limit, please refer to our next reply.*)

---

> ### Author Response · Authors · 2025-11-29
> **The second reply to Reviewer 22Y8**
>
> **2.Line 195: “we compute a representation by averaging two specific last-token representations: the last token of the context (prompt) part and the last token of the label part” is a heuristic design. I can not get the motivation for extracting the tokens from these two positions and averaging them, nor convincing experimental results to justify the necessity of doing so. Specifically, one can expect that the last token of the label already encodes the preceding demonstration information, so extracting only the label token’s hidden states and MLP outputs might suffice. You might conduct more ablation experiments to verify whether extracting both the label token and the demonstration token is truly necessary. (from weakness 1.2)**
>
> Thank you for your insightful comments. We greatly appreciate the opportunity to clarify our design choices. Below is a detailed response.
>
> **(1) Rationale for Multi-Token Representations**
>
> We **introduce the Prompt Last Token (PLT) and Label Last Token (LLT) to address the positional bias inherent in the last-token representations of causal language models (CLMs).** Under the CLMs' autoregressive training objective, the attention distribution of the LLT—the sequence's final token—is disproportionately focused on itself. **This self-referential bias limits the LLT's capacity to effectively integrate information from task instructions and input contexts.** To overcome this limitation, we augment the LLT with a complementary token specifically designed to better capture these contextual signals.
>
> Intuitively, PLT and LLT complement each other to form a compressive representation:
>
> **(i) PLT**: Positioned at the end of the instructions and input text, naturally encodes contextual information (e.g., task requirements, input content), functioning as a **"context encoder"**.
>
> **(ii) LLT**: As the final token of the entire sequence, carries the strongest signal of label semantics, acting as a **"label encoder"**.
>
> By averaging their representations, we fuse the context signal and label signal. This provides the subsequent interaction module with a fused representation that effectively captures both the task context and the corresponding answer pattern.
>
> **(2) Further Validation**
>
> We conducted an ablation study to isolate the impact of multi-token representations, with results shown below (modules added incrementally):
> | Dataset (Avg Len) / Method | SST-2 (246.01) | SST-5 370.31) | TREC (653.63) | AGNews (343.96) | HS18 (512.29) | DBPedia (475.10) | MR (366.94) | Subj (637.94) |
> |--------|-------|-------|------|--------|------|---------|-----|------|
> | I2CL   |90.28  |43.76  |75.44 |84.24   |76.40 |91.87    |85.52|76.36 |
> | *+ Multi-Token* | *91.16 (+0.88)* | *43.86 (+0.10)* | *79.64 (+4.20)* | *84.64 (+0.40)* | *83.36 (+6.96)* | *92.08 (+0.21)* | *86.44 (+0.92)* | *80.60 (+4.24)* |
> | + Demo-Interaction |92.56|44.56|77.16|84.76|82.96|94.48|86.64|77.52|
> | + Demo-Importance |92.32|45.32|86.56|86.44|82.28|93.52|87.00|87.20|
> | + All(I3CL) |92.66|51.16|87.36|87.07|86.28|95.88|90.67|90.84|
>
> **Multi-token representations (+Multi-Token) demonstrate performance gains across all 8 datasets, achieving particularly significant gains on datasets with longer average sequence lengths (e.g., TREC: +4.20, HS18: +6.96, Subj: +4.24).** These findings indicate that the multi-token design effectively mitigates the single LLT's limited context capture in long sequences.
>
> While multi-token representations achieve consistent improvements across all datasets (+2.2% avg), the gains it provides are modest compared to the subsequent gains from the Demo-Interaction and Demo-Importance modules. Crucially, the full I3CL model outperforms all these ablated models, demonstrating that multi-token representations establish an important foundation. This foundation enables expressive interaction to achieve optimal performance.
>
> We will supplement these ablation results and analyses in the revised manuscript.
>
> (*Due to the word limit, please refer to our next reply.*)

---

> ### Author Response · Authors · 2025-11-29
> **The third reply to Reviewer 22Y8**
>
> **3.Line 089: “enabling the model to selectively emphasize the most informative demonstrations” is also not directly supported by experimental evidence. At least, I would like to see a case study of the selected demonstrations by the softmax processing. Also, another concern is raised on this point, see below. (from weakness 1.3)**
>
> Thank you for your valuable suggestion.
>
> We have added a case study to demonstrate how the softmax processing guides the model's selection among possible demonstrations for a given test sample.
>
> Specifically, we selected 6 test samples from the TREC dataset, visualizing the attention weights I3CL allocated to all available demonstrations within a set covering six distinct categories: Person, Number, Description, Location, Abbreviation, and Entity. The attention weight distributions for these samples appear below:
>
> **Case Study: Attention Weights (Post-Softmax) for Demonstrations Across Test Samples**
>
> | Test Samples | Person | Number | Description | Location | Abbreviation | Entity |
> |---|---|---|---|---|---|---|
> | Question: What state has the least amount of rain per year ? Label: Location | 0.136 | **0.170** | 0.165 | 0.168 | 0.154 | 0.167 |
> | Question: Who was the first African American to play for the Brooklyn Dodgers ? Label: Person | **0.181** | 0.166 | 0.162 | 0.164 | 0.171 | 0.165 |
> | Question: What is pilates ? Label: Description | 0.166 | 0.159 | **0.184** | 0.160 | 0.168 | 0.182 |
> | Question: When was the Boston tea party ? Label: Number | 0.139 | **0.188** | 0.169 | 0.182 | 0.155 | 0.167 |
> | Question: What is a mirror made out of ? Label: Entity | 0.165 | 0.167 | 0.167 | 0.166 | 0.165 | **0.169** |
> | Question: What is TUJ ? Label: Abbreviation | 0.169 | 0.166 | 0.165 | 0.162 | **0.172** | 0.166 |
>
> Demonstration Set:
> > - Person: Given Question: What famous husband-and-wife team did radio ads for Blue Nun wine ?\nASSISTANT: The type of the question is person\n
> > - Number: Given Question: What 's men 's par on a 455-yard golf hole ?\nASSISTANT: The type of the question is number\n
> > - Description: Given Question: What function does a community 's water tower serve ?\nASSISTANT: The type of the question is description\n
> > - Location: Given Question: Where is Belize located ?\nASSISTANT: The type of the question is location\n
> > - Abbreviation: Given Question: What does R.E.M. stand for , as in the rock group R.E.M. ?\nASSISTANT: The type of the question is abbreviation\n
> > - Entity: Given Question: What was the name of Roy Rogers 's dog ?\nASSISTANT: The type of the question is entity\n
>
> As shown in the table, attention weights adapt dynamically to each input question. Furthermore, the semantically relevant demonstration receives the highest weight across test samples. This highlights the effectiveness of our importance-weighting mechanism in I3CL.
>
> **4.This essentially reproduces a demonstration selection process, which can benefit not only from the similarity but also from the diversity of demonstrations [1]. Since evaluating diversity does not significantly affect computational efficiency, I suggest that the authors incorporate diversity into their demonstration-to-query interaction process. Furthermore, the authors could even consider parameterizing these softmax processes, since gradient-based optimization is already employed for efficient calibration, adding a few more parameters would be acceptable. (from weakness 2.1)**
>
> Thank you for your insightful comments.
>
> We fully agree that joint consideration of both similarity and diversity in our approach is crucial. This is a promising research direction. We will discuss and cite reference [1] in the "Future Work" section of our paper.
>
> Regarding your suggestion on parameterization, we appreciate exploring this for the softmax functions is worthwhile and plan to investigate it.
>
> (*Due to the word limit, please refer to our next reply.*)

---

> ### Author Response · Authors · 2025-11-29
> **The fourth reply to Reviewer 22Y8**
>
> **5. It is conceivable that, in classification tasks, a demonstration similar to the query is likely to have the same label as the query. Directly copying the hidden states of this consistent label into the zero-shot inference naturally biases the output toward that label, which may cause your method to degenerate into a KNN retriever operating on embedded representations. An intuitive way to address this concern is to conduct experiments on non-classification tasks, as described below. (from weakness 2.2). The proposed method is clearly not limited to classification tasks, yet this paper only evaluated it on classification datasets. (from weakness 3.)**
>
> Thank you for your valuable feedback.
>
> As suggested, we expanded our evaluation to include text style transfer. Following the state-of-the-art implicit ICV method [2], we tested I3CL on the ParaDetox dataset, where the task is to rewrite toxic sentences into neutral expressions while preserving meaning. This requires precise context understanding and controlled generation capabilities. Results are presented in the table below:
>
> | Method                  | Toxicity(%) | ROUGE-1 | BERT Score |
> |-------------------------|-------------|---------|------------|
> | Original Test Set       | 84.58       | -       | -          |
> | Gold-standard Reference | 16.23       | -       | -          |
> | Zero-Shot               | 71.6        | 73.15   | 93.32      |
> | ICL                     | 66.81       | 74.19   | 93.11      |
> | LoRA FT                 | 48.94       | 57.32   | 89.34      |
> | ICV                     | 39.54       | 65.97   | 92.73      |
> | I2CL                    | 57.56       | 70.55   | 90.79      |
> | I3CL(Ours)              | 24.80       | 69.19   | 91.74      |
>
> |Method|Avg Training Loss|
> |------|--------------|
> |I2CL  |2.70|
> |I3CL  |0.67|
>
> **I3CL achieves significantly lower toxicity (24.80%) than all baselines, substantially outperforming even the ICV method (39.54%).** I3CL also maintains strong performance in fluency (ROUGE-1) and semantic preservation (BERT Score), matching or exceeding previous baselines.
>
> Besides, I3CL's improvement over I2CL highlights that the latter's static, coarse-grained context aggregation is insufficient for this generative task. This insufficiency is further evidenced by I2CL's high average training loss (2.70). **In contrast, I3CL's dynamic, interaction-aware mechanism leads to lower average loss of 0.67, enabling superior performance on this generative task.**
>
> **These results demonstrate I3CL's effectiveness on more complex, non-classification tasks.**
>
> We will include these results in the revised manuscript.
>
> (*Due to the word limit, please refer to our next reply.*)

---

> ### Author Response · Authors · 2025-11-29
> **The fifth reply to Reviewer 22Y8**
>
> **6.Moreover, although four models are listed in the paper, this paper conducted a complete evaluation on only one of them. This undermines the credibility of their approach. (from weakness 3.)**
>
> To address this concern, we have supplemented the complete experimental results across the three backbones: LLaMA3-8B, GLM4-9B and DeepSeek-R1-8B. To ensure reliability, all reported results represent the average of 5 independent runs:
>
> **Performance Comparison on LLaMA3-8B**
> | Method       | SST-2 | SST-5 | TREC  | AGNews | HS18 | DBPedia | MR    | Subj  |
> |--------------|-------|-------|-------|--------|------|---------|-------|-------|
> | Zero-Shot    | 88.40 | 47.84 | 67.84 | 70.60  | 63.00 | 78.60  | 82.62 | 71.96 |
> | ICL          | 92.49 | 50.76 | 72.24 | 74.80  | 52.86 | 87.67  | $\underline{\text{87.27}}$ | 74.53 |
> | LoRA-FT      | **94.26** | $\underline{\text{52.48}}$ | $\underline{\text{85.64}}$ | **87.63** | 86.01 | **96.83** | 86.21 | $\underline{\text{91.22}}$ |
> | Noise-Vector | 51.83 | 29.23 | 24.47 | 27.28  | 57.81 | 7.87   | 52.20 | 47.42 |
> | Label-Anchor | 92.45 | 45.86 | 34.62 | 69.84  | 73.93 | 84.28  | 73.75 | 54.83 |
> | Task-Vector  | 91.85 | 45.65 | 54.27 | 85.55  | 80.60 | 67.66  | 82.47 | 53.82 |
> | ICV          | 88.70 | 40.68 | 58.62 | 84.58  | 90.15 | 92.83  | 81.27 | 51.43 |
> | I2CL         | 89.63 | 48.64 | 76.00 | 85.83  | $\underline{\text{90.60}}$ | 93.47  | 82.85 | 77.12 |
> | I3CL         | $\underline{\text{93.43}}$ | **53.60** | **88.28** | $\underline{\text{85.89}}$  | **91.60** | $\underline{\text{94.16}}$  | **88.20** | **92.08** |
>
> **Performance Comparison on GLM4-9B**
> | Method       | SST-2 | SST-5 | TREC  | AGNews | HS18 | DBPedia | MR    | Subj  |
> |--------------|-------|-------|-------|--------|------|---------|-------|-------|
> | Zero-Shot    | 94.20 | 50.84 | 69.60 | 86.21  | 83.43 | 88.83  | 89.28 | 54.04 |
> | ICL          | 95.45 | 51.64 | 77.88 | 86.67  | 84.21 | 97.40  | 89.41 | 61.16 |
> | LoRA-FT      | $\underline{\text{95.61}}$ | $\underline{\text{51.65}}$ | **93.09** | **91.19** | $\underline{\text{85.18}}$ | 97.85 | **91.40** | $\underline{\text{82.21}}$ |
> | Noise-Vector | 48.27 | 23.64 | 18.88 | 25.81  | 47.82 | 7.81   | 54.62 | 55.18 |
> | Label-Anchor | 90.88 | 49.46 | 53.43 | 48.41  | 71.64 | 83.29  | 81.04 | 53.80 |
> | Task-Vector  | 95.85 | 48.42 | 51.27 | 68.64  | 74.41 | 70.87  | 82.25 | 62.80 |
> | ICV          | 94.36 | 49.68 | 69.20 | 84.46  | 80.81 | 96.21  | 86.83 | 63.27 |
> | I2CL         | 94.63 | 51.00 | 77.84 | 88.60  | 81.49 | 97.86  | 88.10 | 59.60 |
> | I3CL         | **95.92** | **52.36** | $\underline{\text{92.76}}$ | $\underline{\text{89.66}}$  | **87.81** | **98.24** | $\underline{\text{90.04}}$ | **82.40** |
>
> **Performance Comparison on DeepSeek-R1-8B**
> | Method       | SST-2 | SST-5 | TREC  | AGNews | HS18 | DBPedia | MR    | Subj  |
> |--------------|-------|-------|-------|--------|------|---------|-------|-------|
> | Zero-Shot    | 86.63 | 30.80 | 45.40 | 80.21  | 80.27 | 87.31  | 73.81 | 53.64 |
> | ICL          | 91.10 | 39.28 | 68.40 | 81.22  | 82.26 | 90.80  | 85.25 | 55.56 |
> | LoRA-FT      | $\underline{\text{91.86}}$ | $\underline{\text{47.51}}$ | **90.39** | $\underline{\text{88.20}}$ | $\underline{\text{89.30}}$ | **97.82**  | **86.85** | $\underline{\text{64.83}}$ |
> | Noise-Vector | 51.80 | 19.83 | 19.41 | 26.86  | 51.23 | 8.28   | 52.37 | 52.65 |
> | Label-Anchor | 86.46 | 45.84 | 54.29 | 66.49  | 56.48 | 75.65  | 85.87 | 50.51 |
> | Task-Vector  | 86.53 | 29.86 | 66.80 | 83.45  | 83.68 | 64.97  | 83.46 | 47.35 |
> | ICV          | 87.88 | 28.02 | 41.87 | 80.87  | 85.04 | 81.87  | 81.82 | 63.27 |
> | I2CL         | 81.84 | 33.00 | 73.26 | 84.63  | 86.26 | 88.64  | 80.82 | 61.64 |
> | I3CL         | **91.91** | **49.55** | $\underline{\text{89.60}}$ | **89.69** | **90.74** | $\underline{\text{96.45}}$ | $\underline{\text{86.46}}$ | **76.00** |
>
> In conclusion, **I3CL delivers consistent and significant improvements across all evaluated backbones, reinforcing its effectiveness as a generalizable framework.** These results will be included in the revised manuscript.
>
> (*Due to the word limit, please refer to our next reply.*)

---

> ### Author Response · Authors · 2025-11-29
> **The sixth reply to Reviewer 22Y8**
>
> **7. This work is incremental. Although I believe the improvements made in this paper are reasonable, it still largely follows the framework of Li et al. Personally, I do not object to incremental work, but I am unsure whether it fits the standards of ICLR. I therefore raise this point for the AC’s consideration, though it does not affect my main score. (from weakness 4.)**
>
> Thank you for your time in reviewing our paper and for your valuable feedback. We appreciate the opportunity to further clarify our research's core contributions.
>
> We argue that the field of implicit In-Context Learning (ICL) faces a pivotal trade-off: **methods pursuing heightened inference efficiency (e.g., I2CL) sacrifice interactive capabilities crucial to standard ICL, such as demo-demo and demo-query interactions.** However, a substantial body of research demonstrates these interactions are fundamental for models' reasoning capacities.
>
> Therefore, the core objective of our work, I3CL, is to address this fundamental imbalance in the field. While building upon I2CL framework, I3CL introduces interaction mechanisms that represent a significant conceptual departure: shifting from the traditional paradigm of **'sacrificing performance for efficiency'** towards **'recovering essential capabilities while preserving efficiency'**. Our lightweight approach, as extensive experimental results demonstrate, **delivers a significant average performance gain of 7.26% with only minimal computational overhead.** This strongly suggests that the restored interactive capabilities are essential for advancing implicit ICL.
>
> Our work addresses critical limitations in the field and demonstrates significant performance gains, reflecting ICLR's mission to advance understanding at the frontiers of the discipline. We thank you again for your thorough review and consideration.
>
> **8. Table 5 (Fig. 5 in your notation): As far as I understand, compared with I3CL, each step in I2CL uses a simpler module variant (e.g., average pooling instead of self-attention). Why, then, is its time cost greater than that of I3CL? (from question 1.)**
>
> Thank you for raising this important point. We agree that, in principle, the simpler I2CL module should be more efficient than I3CL.
>
> **The unexpectedly higher time cost initially reported for I2CL resulted from an inconsistency in the experiment setup**: specifically, I2CL used a batch size of 1 while I3CL used 2. We apologize for this error.
>
> **To ensure a fair comparison, we standardized the batch size to 1 across all models and systematically retested their efficiency.** This revised analysis, extended to include multi-shot scenarios beyond the initial few-shot tests. The detailed results are as follows:
>
> **Calibration Time Comparison**
>
> | Method |Acc| 5-Shot | 10-Shot | 30-Shot | 100-Shot |
> |--------|----|-----------|--------|---------|---------|
> | I2CL |75.44| 1x | 1x | 1x | 1x |
> | I3CL (Ours) |87.36| 1.02x | 1.03x | 1.07x | 1.34x |
>
> **Inference Time Comparison**
>
> | Method |Acc| 5-Shot | 10-Shot | 30-Shot | 100-Shot |
> |--------|-|-----------|--------|---------|---------|
> |Zero-Shot$_\text{bs=1}$|41.96|1x|1x|1x|1x|
> | ICL$_\text{bs=1}$ |60.08| 5.84x | 9.93x | 36.66x | - |
> |I2CL$_{\text{bs=1}}$|75.44|4.08|4.08|4.08|4.08|
> | I3CL$_{\text{bs=1}}$ |87.36| 4.10x | 4.13x | 4.27x | 4.91x |
> | I3CL (Ours)$_\text{bs=2}$ |87.36| 3.94x | - | -| - |
>
> **Peak Memory Usage**
>
> | Method |Acc| 5-Shot | 10-Shot | 30-Shot | 100-Shot |
> |--------|-|-----------|--------|---------|---------|
> |Zero-Shot|41.96|1x|1x|1x|1x|
> | ICL |60.08| 1.59x | 2.15x | 5.84 | - (OOM) |
> | I2CL |75.44| 2.04x | 2.04x | 2.04x | 2.04x |
> | I3CL (Ours) |87.36| 2.15x | 2.25x | 2.45x | 2.83x |
>
> We make three key observations：
>
> **(1) Calibration Efficiency:** In standard few-shot settings (5-30 shots), I3CL adds <=7% overhead compared to I2CL, demonstrating comparable efficiency. While 100-shot calibration increases by 34% (reflecting interactive complexity), this one-time offline cost is justified by  significant accuracy gains (+11.92%).
>
> **(2) Inference Speed:** In critical few-shot scenarios (5-30 shots) show <=1.3% increase in inference latency. Although overhead rises at 100-shot (4.91x vs. I2CL's 4.08x), it remains dramatically lower than standard ICL.
>
> **(3) Memory Efficiency:** I3CL’s peak memory usage closely matches I2CL's. Crucially, I3CL avoids Out-Of-Memory (OOM) errors in all tested scenarios, unlike standard ICL which fails at 100-shot. This demonstrating I3CL’s superior scalability.
>
> **To ensure full reproducibility, we will publicly release all relevant code.** These results and discussions will be included in the revised manuscript.
>
> (*Due to the word limit, please refer to our next reply.*)

---

> ### Author Response · Authors · 2025-11-29
> **The seventh reply to Reviewer 22Y8**
>
> **9. Line 218: I did not fully understand what “global collective regularities” specifically refers to. I can imagine that you might be referring to something similar to a “task representation”, but I would like to know what these “regularities” actually are in numerical terms. Therefore, you may need to provide a more detailed analysis comparing the task vector extracted by I2CL with that of I3CL, highlighting their semantic differences. (from question 2.)**
>
> Thank you for this insightful suggestion.
>
> Your understanding is correct. 'Global collective regularities' denotes a universal, task-level vector representation. Specifically, it is the property that the context vector extracted from any arbitrary set of demonstrations for a given task should exhibit:
>
> **(1) High Intra-Task Consistency:** Capturing the task’s most essential and stable features, resulting in context vectors that remain highly similar across varied demonstration examples.
>
> **(2) Clear Inter-Task Distinction:** Filters out superficial or task-irrelevant features, ensuring context vectors for different tasks exhibit distinct separation.
>
> To directly measure these properties, we sampled distinct demonstration sets from two separate datasets (A and B). We then computed the cosine similarity for all possible task combinations. Results for both I2CL and I3CL are shown below; bold diagonal values indicate intra-task similarity.
>
>
> **I2CL: Task Vector Similarity Across Different Demonstration Sets**
> |Set A\Set B| SST-2 | SST-5 | TREC | AGNews | HS18 | DBPedia | MR | Subj |
> |-------|-------|-------|-------|--------|-------|---------|-------|-------|
> | SST-2 | **98.91** | 97.54 | 85.21 | 86.42 | 87.47 | 85.44 | 98.49 | 91.27 |
> | SST-5 | 96.00 | **99.84** | 85.36 | 88.04 | 88.86 | 87.41 | 98.30 | 92.07 |
> | TREC | 85.92 | 85.42 | **99.91** | 86.69 | 79.89 | 88.07 | 83.50 | 83.55 |
> | AGNews | 84.78 | 87.05 | 85.51 | **99.47** | 82.27 | 86.83 | 86.36 | 87.00 |
> | HS18 | 85.85 | 88.62 | 80.09 | 82.41 | **99.52** | 83.59 | 87.87 | 91.37 |
> | DBPedia | 85.35 | 87.56 | 87.95 | 87.45 | 83.81 | **99.67** | 85.59 | 85.71 |
> | MR | 97.19 | 98.39 | 84.24 | 86.88 | 88.25 | 85.67 | **99.60** | 91.85 |
> | Subj | 89.65 | 91.48 | 83.72 | 87.78 | 90.60 | 85.49 | 91.63 | **99.35** |
>
>
> **I3CL: Task Vector Similarity Across Different Demonstration Sets**
> |Set A\Set B| SST-2 | SST-5 | TREC | AGNews | HS18 | DBPedia | MR | Subj |
> |-------|-------|-------|--------|--------|-------|---------|-------|--------|
> | SST-2 | **99.99** | 65.40 | -0.5049 | 65.78 | 67.51 | 61.86 | 97.88 | 71.74 |
> | SST-5 | 67.30 | **95.43** | 16.36 | 57.74 | 59.25 | 55.41 | 68.15 | 58.52 |
> | TREC | -0.5774 | 17.59 | **99.92** | 0.5215 | 0.4349 | 0.8899 | -0.5703 | 0.1354 |
> | AGNews | 65.59 | 56.40 | 0.5033 | **98.41** | 63.96 | 66.90 | 66.97 | 67.21 |
> | HS18 | 67.37 | 57.45 | 0.4607 | 63.89 | **97.33** | 56.63 | 68.41 | 70.67 |
> | DBPedia | 62.23 | 55.04 | 0.8938 | 66.96 | 56.95 | **99.36** | 63.16 | 60.09 |
> | MR | 97.88 | 66.24 | -0.4971 | 67.07 | 68.54 | 62.75 | **98.99** | 72.97 |
> | Subj | 71.77 | 57.47 | 0.1954 | 67.06 | 70.97 | 59.94 | 73.03 | **97.05** |
>
> **(1) I2CL produces indistinct task representations**, evidenced by its high inter-task similarity (typically exceeding 85%) alongside its high intra-task consistency (>98% diagonal). This indicates significant contamination by task-irrelevant features, hindering clear task separation. This is notably observed in the high SST-2/TREC similarity (85.21%).
>
> **(2) In contrast, I3CL captures distinct task-specific patterns,** maintaining strong intra-task consistency (>95% diagonal) while achieving sharp inter-task discrimination. For unrelated tasks (e.g., SST-2 vs. TREC), similarity drops to near-zero levels. For related tasks (e.g., SST-2 vs. SST-5) with differing classification goal, similarity remains moderated (65.40%), significantly lower than I2CL’s 97.54%. This underscores I3CL's ability to encode tasks with greater nuance.
>
> **These results demonstrate that I3CL learns highly discriminative task representations, capturing underlying global regularities.** This precise encoding underscores I3CL's representational superiority over I2CL.
>
> (*Due to the word limit, please refer to our next reply.*)

---

> ### Author Response · Authors · 2025-11-29
> **The eighth reply to Reviewer 22Y8**
>
> **10. Line 414: What does the “highest weight” refer to? Is the weight the norm of the hidden state? Line 456: What does “noise” refer to here? (from question 3. and 4.)**
>
> Thank you for your questions. We provide detailed explanations below and will include them in the revised manuscript.
>
> **(1) Meaning of "Highest Weight" (Line 414):**
> "The weight" refers to the logits vector before softmax normalization. "The highest weight" means the maximum value within this logits vector. A higher value indicates greater model confidence or certainty that this specific token is the correct next prediction.
>
> **(2) Meaning of "Noise" (Line 456):**
> "Noise" refers to standard Gaussian noise vectors injected to verify the method's robustness. Specifically, we added these noise vectors to the context vectors at varying magnitudes. This addition simulated the potential presence of irrelevant noise within the demonstration set relative to the current query. This experiment demonstrates that I3CL effectively suppresses the influence of such noise.
>
> **11.Line 617: Typo. (from question 5.)**
>
> Thank you for your careful review and for catching this typo. We have removed the extra period at the end of line 617 in the revised manuscript.
>
> **References**
>
> [1] Mavromatis C et al. Which Examples to Annotate for In-Context Learning? Towards Effective and Efficient Selection, ICLR 2024.
>
> [2] Liu S et al. In-context vectors: making in context learning more effective and controllable through latent space steering, ICML 2024.

---

### Official Review · Reviewer_xuZX · 2025-10-25

**Soundness:** 2
**Presentation:** 3
**Contribution:** 2
**Rating:** 6
**Confidence:** 4

**Summary:**

The paper introduces Interactive Implicit In-Context Learning (I3CL), a new framework designed to improve upon existing implicit in-context learning (I2CL) methods. The authors identify two weaknesses of I2CL: 1) I2CL processes each demonstration in isolation, ignoring the valuable relationships between demonstrations; 2) It uses simple average pooling to create the task vector, treating all demonstrations as equally informative for any given query. I3CL solves these problems by introducing two lightweight attention mechanisms for 1) inter-demonstration interaction and 2) demonstration-query interaction, to improve the analytical capabilities of I2CL.

I3CL weighs the in-context vectors and injects them into the LLM's latent space using learnable weights to guide the final prediction. Experiments are conducted on classification tasks and various LLMs, showing that I3CL significantly outperforms I2CL and even surpasses the performance of fine-tuned LoRA models, while maintaining comparable inference efficiency to I2CL.

**Strengths:**

- The problem background, motivation, and formulation are clear and easy to understand.
- The paper identifies an interesting problem, connecting the literature on importance attribution of in-context demonstrations and in-context vectors to form an elegant solution.
- The method is grounded on well-defined mathematical formulations, showing its rigor and interpretability.
- There is some empirical backing of the superiority of the proposed method across datasets and LLMs.

**Weaknesses:**

- The datasets are somewhat limited in scope. All experiments are conducted on text-classification tasks, which is too simplistic. It would be great to see experiments on QA tasks using expert trajectories as demonstrations, similar to [1]. Or, at least some non-classification tasks, for example used in [4]
- While it is interesting to connect importance weighting of in-context demonstrations with I2CL, the literature review lacks references to previous works in importance weighting of in-context demonstrations, for example [2,3].
- I hope the authors provide a more detailed explanation of how to interpret Eq. 6 as the "importance weight" of in-context vectors.
-  As far as I understand the paper, Eq. 8 and Eq. 9 follow from Eq. 4 and Eq. 5 of the I2CL paper. I encourage the authors to properly reference the paper when introducing these two equations.



[1] Monea et al., LLMs Are In-Context Bandit Reinforcement Learners, COLM 2025.

[2] Zhou et al., DETAIL: Task Demonstration Attribution for Interpretable In-context Learning, NeurIPS 2024.

[3] Yang et al., Representative Demonstration Selection for In-Context Learning with Two-Stage Determinantal Point Process, EMNLP 2023.

[4] Todd et al., Function Vectors in Large Language Models, ICLR 2024.

**Questions:**

- The calibration process seems to require some task demonstrations for training. Does the calibration process take place during inference or is it conducted during a "boostrapping" stage?
- The implementation of I3CL seems to be more complicated than I2CL. I wonder if the authors can provide some explanation for why the inference speed is "near-parity". It would be helpful if the authors could offer some speed comparison between I3CL and I2CL.
- For tasks that require the model to generate multiple tokens, do the in-context vectors need to be injected at every autoregressive decoding step?


I am open to raising the score should the authors sufficiently address my concerns.

---

> ### Author Response · Authors · 2025-11-29
> **The first reply to Reviewer xuZX**
>
> We sincerely thank you for taking the time to review our work!
>
> **1. The datasets are somewhat limited in scope. All experiments are conducted on text-classification tasks, which is too simplistic. It would be great to see experiments on QA tasks using expert trajectories as demonstrations, similar to [1]. Or, at least some non-classification tasks, for example used in [4]. (from weakness 1.)**
>
> Thank you for your valuable feedback.
>
> **As suggested, we expanded our evaluation to include text style transfer task.** Following the state-of-the-art implicit ICV method [5], we tested I3CL on the ParaDetox dataset, where the task is to rewrite toxic sentences into neutral expressions while preserving meaning. This requires precise context understanding and controlled generation capabilities. Results are presented in the table below:
>
> | Method                  | Toxicity(%) | ROUGE-1 | BERT Score |
> |-------------------------|-------------|---------|------------|
> | Original Test Set       | 84.58       | -       | -          |
> | Gold-standard Reference | 16.23       | -       | -          |
> | Zero-Shot               | 71.6        | 73.15   | 93.32      |
> | ICL                     | 66.81       | 74.19   | 93.11      |
> | LoRA FT                 | 48.94       | 57.32   | 89.34      |
> | ICV                     | 39.54       | 65.97   | 92.73      |
> | I2CL                    | 57.56       | 70.55   | 90.79      |
> | I3CL(Ours)              | 24.80       | 69.19   | 91.74      |
>
> |Method|Avg Training Loss|
> |------|--------------|
> |I2CL  |2.70|
> |I3CL  |0.67|
>
> **I3CL achieves significantly lower toxicity (24.80%) than all baselines, substantially outperforming even the ICV method (39.54%).** I3CL also maintains strong performance in fluency (ROUGE-1) and semantic preservation (BERT Score), matching or exceeding previous baselines.
>
> Besides, I3CL's improvement over I2CL highlights that the latter's static, coarse-grained context aggregation is insufficient for this generative task. This insufficiency is further evidenced by I2CL's high average training loss (2.70). **In contrast, I3CL's dynamic, interaction-aware mechanism leads to lower average loss of 0.67, enabling superior performance on this generative task.**
>
> **These results demonstrate I3CL's effectiveness on more complex, non-classification tasks.**
>
> We will include these results in the revised manuscript.
>
> **2. While it is interesting to connect importance weighting of in-context demonstrations with I2CL, the literature review lacks references to previous works in importance weighting of in-context demonstrations, for example [2,3]. (from weakness 2.)**
>
> Thank you for this valuable suggestion. We fully agree that discussing prior work on importance weighting in in-context demonstrations, specifically including references like [2,3], would enhance the depth of our Related Work section.
>
> To address this, we will add a paragraph to the Related Work section detailing contributions from papers [2,3] alongside other key works in this area.
>
> **3. I hope the authors provide a more detailed explanation of how to interpret Eq. 6 as the "importance weight" of in-context vectors. (from weakness 3.)**
>
> Thank you for this valuable suggestion.
>
> To address this point, we now provide a detailed explanation of how Eq. 6 produces the importance weight for in-context vectors.
>
> Specifically, consider the score term $\text{score}^l_{a,x_t}$ in Eq. 6 as an example:
>
> $\text{score}^l_{a,x_t} = \text{softmax}(a_{x_t}^l \cdot c'^{l\top}_{a})$
>
> Overall, this equation derives normalized importance weights that quantify the relevance of each in-context demonstration relative to each token in the test sample at layer $l$. Specifically, this calculation involves:
>
> Query($a_{x_t}^l \in \mathbb{R}^{s \times d}$): The representation of the test sample $x_t$ transformed at layer $l$, where $s$ is sequence length and $d$ is embedding dimension.
>
> Key($c'^{l}_{a} \in \mathbb{R}^{|\mathcal{D}| \times d}$): The aggregated representations of demonstration set $\mathcal{D}$, forming a semantic knowledge base at layer at layer $l$.
>
> The dot product $a^l_{x_t} \cdot c'^{l\top}_{a}$ computes pairwise token-demonstration similarities. These raw similarities are normalized via $\text{softmax}$ applied along the demonstration axis ($|\mathcal{D}|$), yielding the matrix
>
> $\text{score}^l_{a,x_t} \in \mathbb{R}^{s \times |\mathcal{D}|}$. Each element $\text{score}^l_{a,x_t}[i, j]$ quantifies the relevance of the $j$-th demonstration to the $i$-th token in $x_t$.
>
> We will add these explanations to improve manuscript clarity.
>
> (*Due to the word limit, please refer to our next reply.*)

---

> ### Author Response · Authors · 2025-11-29
> **The second reply to Reviewer xuZX**
>
> **4. As far as I understand the paper, Eq. 8 and Eq. 9 follow from Eq. 4 and Eq. 5 of the I2CL paper. I encourage the authors to properly reference the paper when introducing these two equations. (from weakness 4.)**
>
> Thank you for your careful review.
>
> In the revised manuscript, we will explicitly indicate that Equations (8) and (9) are derived from Equations (4) and (5) of the I2CL paper, and will add a reference to the source upon their first appearance in the text.
>
>
> **5. The calibration process seems to require some task demonstrations for training. Does the calibration process take place during inference or is it conducted during a "boostrapping" stage? (from question 1.)**
>
> Thank you for your insightful question.
>
> Yes, calibration is performed using a task-specific demonstration set. The fusion coefficients ($\lambda$ and $\beta$) are calibrated offline via this data in a bootstrapping training stage.  Once calibrated, these coefficients remain fixed during inference.
>
>
> **6.The implementation of I3CL seems to be more complicated than I2CL. I wonder if the authors can provide some explanation for why the inference speed is "near-parity". It would be helpful if the authors could offer some speed comparison between I3CL and I2CL. (from question 2.)**
>
> Thank you for this valuable suggestion.
>
> Below, we present an extensive efficiency analysis comparing I3CL and I2CL costs, covering calibration time, inference time, and peak memory usage. All timings are normalized against the fastest baseline in each category, with 1x denoting the reference value.
>
> **Calibration Time Comparison**
>
> | Method |Acc| 5-Shot | 10-Shot | 30-Shot | 100-Shot |
> |--------|----|-----------|--------|---------|---------|
> | I2CL |75.44| 1x | 1x | 1x | 1x |
> | I3CL (Ours) |87.36| 1.02x | 1.03x | 1.07x | 1.34x |
>
> **Inference Time Comparison**
>
> | Method |Acc| 5-Shot | 10-Shot | 30-Shot | 100-Shot |
> |--------|-|-----------|--------|---------|---------|
> |Zero-Shot$_\text{bs=1}$|41.96|1x|1x|1x|1x|
> | ICL$_\text{bs=1}$ |60.08| 5.84x | 9.93x | 36.66x | - |
> |I2CL$_{\text{bs=1}}$|75.44|4.08|4.08|4.08|4.08|
> | I3CL$_{\text{bs=1}}$ |87.36| 4.10x | 4.13x | 4.27x | 4.91x |
> | I3CL (Ours)$_\text{bs=2}$ |87.36| 3.94x | - | -| - |
>
> **Peak Memory Usage**
>
> | Method |Acc| 5-Shot | 10-Shot | 30-Shot | 100-Shot |
> |--------|-|-----------|--------|---------|---------|
> |Zero-Shot|41.96|1x|1x|1x|1x|
> | ICL |60.08| 1.59x | 2.15x | 5.84 | - (OOM) |
> | I2CL |75.44| 2.04x | 2.04x | 2.04x | 2.04x |
> | I3CL (Ours) |87.36| 2.15x | 2.25x | 2.45x | 2.83x |
>
> We make three key observations：
>
> **(1) Calibration Efficiency:** In standard few-shot settings (5-30 shots), I3CL adds <=7% overhead compared to I2CL, demonstrating comparable efficiency. While 100-shot calibration increases by 34% (reflecting interactive complexity), this one-time offline cost is justified by  significant accuracy gains (+11.92%).
>
> **(2) Inference Speed:** In critical few-shot scenarios (5-30 shots) show <=1.3% increase in inference latency. Although overhead rises at 100-shot (4.91x vs. I2CL's 4.08x), it remains dramatically lower than standard ICL.
>
> **(3) Memory Efficiency:** I3CL’s peak memory usage closely matches I2CL's. Crucially, I3CL avoids Out-Of-Memory (OOM) errors in all tested scenarios, unlike standard ICL which fails at 100-shot. This demonstrates I3CL’s superior scalability.
>
> In our original manuscript, the 5-shot inference times (I2CL: 4.08x; I3CL: 3.96x) were incomparable due to inconsistent batch sizes (1 vs. 2). As shown in the table above, we have resolved this discrepancy by standardizing all models to batch size 1. We sincerely thank Reviewer #22Y8 for identifying this issue.
>
> **7. For tasks that require the model to generate multiple tokens, do the in-context vectors need to be injected at every autoregressive decoding step? ( from question 3.)**
>
> Thank you for this insightful question.
>
> Yes, our approach injects contextual vectors at each decoding step. This ensures every generated token benefits from the in-context guidance, preserving coherence throughout long-form generation.
>
> **References**
>
> [1] Monea et al., LLMs Are In-Context Bandit Reinforcement Learners, COLM 2025.
>
> [2] Zhou et al., DETAIL: Task Demonstration Attribution for Interpretable In-context Learning, NeurIPS 2024.
>
> [3] Yang et al., Representative Demonstration Selection for In-Context Learning with Two-Stage Determinantal Point Process, EMNLP 2023.
>
> [4] Todd et al., Function Vectors in Large Language Models, ICLR 2024.
>
> [5] Liu S et al. In-context vectors: making in context learning more effective and controllable through latent space steering, ICML 2024

---

### Official Review · Reviewer_X6BH · 2025-11-09

**Soundness:** 3
**Presentation:** 3
**Contribution:** 3
**Rating:** 6
**Confidence:** 4

**Summary:**

This paper introduces Interactive Implicit In-Context Learning (I3CL), a framework that enhances Implicit In-Context Learning (I2CL) by incorporating two lightweight attention mechanisms. The first is inter-demonstration interaction during encoding, which allows allows demonstrations to exchange and contrast information. The second is demonstration-to-query interaction during inference, which dynamically weights demonstrations according to their relevance to the query. Experiments in this paper show that I3CL consistently outperforms existing baselines.

**Strengths:**

1. Clear motivation and well-articulated weaknesses of I2CL: This paper points out that independent encoding and uniform averaging used in I2CL obscure relational and relevance information between demonstrations, providing an intuitive and well-founded motivation for I3CL.

2. Simple and intuitively reasonable design: The addition of inter- and cross-attention modules is conceptually simple, lightweight, and easy to integrate with existing I2CL pipelines.

3. Comprehensive and strong empirical results: Experiments in this paper show that I3CL consistently outperforms existing baselines.

**Weaknesses:**

1. Although this paper provides clear figures that effectively illustrate the high-level idea of their method, the technical details still require more precise explanations. For example, in lines 211–214, the authors mention adopting a self-attention mechanism, but the exact formulation of this mechanism is not provided — are there any trainable parameters involved? In addition, the loss function defined in Section 3.4 and the procedure for training the learnable coefficients could also be described in greater detail.

2. In lines 183–186, each demonstration requires the additional task instruction I. Is this design necessary? Could it be redundant or repetitive, and would it cause significant additional computational or memory overhead when the number of demonstrations becomes large?

**Questions:**

1. What's the exact formulation of the self-attention mechanism in lines 211–214? Are there any trainable parameters involved? If there are  trainable parameters, how are they trained?

2. In lines 192-205, the authors mentioned that they used two specific last-token representations while prior works only used one. Why choose these two specific tokens? Could the authors provide some explanation? Is the improved performance of authors' method compared to previous approaches partly due to the additional two token representations?

---

> ### Author Response · Authors · 2025-11-29
> **The first reply to Reviewer X6BH.**
>
> We sincerely thank you for taking the time to review our work!
>
> **1. Although this paper provides clear figures that effectively illustrate the high-level idea of their method, the technical details still require more precise explanations. For example, in lines 211–214, the authors mention adopting a self-attention mechanism, but the exact formulation of this mechanism is not provided — are there any trainable parameters involved? In addition, the loss function defined in Section 3.4 and the procedure for training the learnable coefficients could also be described in greater detail. (from weakness 1. and question 1.)**
>
> Thank you for your careful review. Below, we provide detailed answers to your questions:
>
> **(1) Self-Attention Mechanism (Lines 211-214)**
>
> The self-attention mechanism follows the standard scaled dot-product formulation [1] with trainable parameters. For the $l$-th layer’s context vector $C^l \in \mathbb{R}^{|D| \times 2 \times d}$, we first project it into Query (Q), Key (K), and Value (V) vectors using trainable weight matrices $W_Q,W_K,W_V\in \mathbb{R}^{d\times d}$. The self-attention output is then computed as:
>
> $$C'^l=\text{Softmax}(\frac{Q\cdot K^T}{\sqrt{d}})\cdot V,$$
>
> where $Q=C^l W_Q, K=C^l W_K, V=C^l W_V$.
>
> **(2) Loss Function in Section 3.4**
>
> We optimize the model to minimize the negative log-likelihood (NLL) of labels $y$ across the demonstration set $\mathcal{D}$. The loss function is defined as:
>
> $$\mathcal{L}=-\frac{1}{|\mathcal{D}|}\sum_{(x,y)\in\mathcal{D}}{\rm log} P(y|x,\lambda,\beta,C^*_{x}),$$
>
> where $P(y|x,\lambda,\beta,C^*_{x})$ denotes the LLM’s predictive probability of label $y$ for the test sample $x$; $C^\*_{x}$ denotes the context vector of the test sample; $\lambda$ and $ \beta$ denote the learnable coefficients.
>
> **(3) Training of Learnable Coefficients**
>
> To optimize the coefficients, we employ a parameter-efficient fine-tuning (PEFT) strategy. Specifically, we freeze all pre-trained parameters of the LLM and update only two sets of parameters: (1) parameters of the interaction modules (e.g., the weight matrices $W_Q,W_K,W_V\in \mathbb{R}^{d\times d}$ in self-attention), and (2) the coefficients $\lambda$ and $\beta$. All parameters are updated using the AdamW optimizer via gradient descent.
> In the revised manuscript, we will add these explanations to enhance clarity.
>
> **2. In lines 183–186, each demonstration requires the additional task instruction I. Is this design necessary? Could it be redundant or repetitive, and would it cause significant additional computational or memory overhead when the number of demonstrations becomes large? (from weakness 2.)**
>
> Thank you for your insightful comments.
>
> This design is to preserve the semantic completeness of each demonstration. Encoding a demonstration in isolation requires the instruction $I$ to provide critical context: the task purpose and the input/output format. Without $I$, the LLM can struggle to connect the demonstration's content to the task goal.
>
> To validate the necessity, we conducted an ablation study on this design. The results are presented in the table below:
>
> | Method              | SST-2 | SST-5 | TREC | AGNews | HS18 | DBPedia | MR   | Subj | Avg.  |
> |---------------------|-------|-------|------|--------|------|---------|------|------|-------|
> | I3CL                | 92.66 | 51.16 | 87.36| 87.07  | 86.28| 95.88   | 90.67| 90.84| 85.24 |
> | w/o Demo-Instruction| 91.20 | 49.00 | 86.40| 85.68  | 84.20| 93.40   | 87.82| 82.68| 82.55 |
>
> **Removing per-demonstration instructions (w/o Demo-Instruction) consistently degrades performance, with average accuracy dropping from 85.24% to 82.55%.** This result underscores the important role of instruction $I$ in enabling the model to extract high-quality representations from each demonstration.
>
> **We acknowledge that repeated encoding of task instructions can introduce extra cost. However, in our approach, each demonstration is encoded only once, independently.** These representations are then cached and reused throughout inference. This caching ensures that the computational overhead scales linearly with the number of demonstrations, maintaining efficiency as the number increases.
>
> *(Due to the word limit, please refer to our next reply.)*

---

> ### Author Response · Authors · 2025-11-29
> **The second reply to Reviewer X6BH**
>
> **3. In lines 192-205, the authors mentioned that they used two specific last-token representations while prior works only used one. Why choose these two specific tokens? Could the authors provide some explanation? Is the improved performance of authors' method compared to previous approaches partly due to the additional two token representations? (from question 2.)**
>
>
> Thank you for your insightful questions. We appreciate the opportunity to clarify our design choices. Below is a detailed explanation.
>
> **(1) Rationale for Multi-Token Representations**
>
> We introduce the Prompt Last Token (PLT) and Label Last Token (LLT) to **address the positional bias inherent in the last-token representations of causal language models (CLMs).** Under the CLMs' autoregressive training objective, the attention distribution of the LLT—the sequence's final token—is disproportionately focused on itself. **This self-referential bias limits the LLT's capacity to effectively integrate information from task instructions and input contexts.** To overcome this limitation, we augment the LLT with a complementary token specifically designed to better capture these contextual signals.
>
> Intuitively, PLT and LLT complement each other to form a compressive representation:
>
> **(i) PLT**: Positioned at the end of the instructions and input text, naturally encodes contextual information (e.g., task requirements, input content), functioning as a **"context encoder"**.
>
> **(ii) LLT**: As the final token of the entire sequence, carries the strongest signal of label semantics, acting as a **"label encoder"**.
>
> By averaging their representations, we fuse the context signal and label signal. This provides the subsequent interaction module with a fused representation that effectively captures both the task context and the corresponding answer pattern.
>
> **(2) Further Validation**
>
> We conducted an ablation study to isolate the impact of multi-token representations, with results shown below (modules added incrementally):
> | Dataset (Avg Len) / Method | SST-2 (246.01) | SST-5 (370.31) | TREC (653.63) | AGNews (343.96) | HS18 (512.29) | DBPedia (475.10) | MR (366.94) | Subj (637.94) |
> |--------|-------|-------|------|--------|------|---------|-----|------|
> | I2CL   |90.28  |43.76  |75.44 |84.24   |76.40 |91.87    |85.52|76.36 |
> | *+ Multi-Token* | *91.16 (+0.88)* | *43.86 (+0.10)* | *79.64 (+4.20)* | *84.64 (+0.40)* | *83.36 (+6.96)* | *92.08 (+0.21)* | *86.44 (+0.92)* | *80.60 (+4.24)* |
> | + Demo-Interaction |92.56|44.56|77.16|84.76|82.96|94.48|86.64|77.52|
> | + Demo-Importance |92.32|45.32|86.56|86.44|82.28|93.52|87.00|87.20|
> | + All(I3CL) |92.66|51.16|87.36|87.07|86.28|95.88|90.67|90.84|
>
> Multi-token representations (+Multi-Token) demonstrate performance gains across all 8 datasets, achieving particularly significant gains on datasets with longer average sequence lengths (e.g., TREC: +4.20, HS18: +6.96, Subj: +4.24). These findings indicate that **the multi-token design effectively mitigates the single LLT's limited context capture in long sequences**.
>
> **While multi-token representations achieve consistent improvements across all datasets (+2.2% avg), the gains it provides are modest compared to the subsequent gains from the Demo-Interaction and Demo-Importance modules.** Crucially, the full I3CL model outperforms all these ablated models, demonstrating that multi-token representations establish an important foundation. This foundation enables expressive interaction to achieve optimal performance.
>
> We will supplement these ablation results and analyses in the revised manuscript.
>
> **References**
>
> [1] Vaswani A et al. Attention is all you need. NeurIPS 2017.

---

### Author Response · Authors · 2025-12-03
**Brief Summary for the Area Chair**

**Dear AC,**

Thank you for your time and oversight. We are grateful to the reviewers for their insightful comments. We have provided comprehensive responses to all points and have revised the manuscript accordingly. Below is a concise summary of our key revisions and responses to the major concerns.

---

## 1. Core Contribution & Clarified Motivation

Our work, **I3CL**, addresses a critical trade-off in implicit in-context learning: efficient methods like I2CL sacrifice the vital inter-demonstration and demo-query interactions inherent in standard ICL. We demonstrate that introducing lightweight, learnable interaction modules can restore and enhance this capability with minimal inference overhead (<1.3% latency increase), leading to an average performance gain of **+7.26%** across multiple tasks and model backbones. This shifts the paradigm from: **"sacrificing performance for efficiency"** $\rightarrow$ **"recovering essential capabilities while preserving efficiency"**.

---

## 2. Substantive Responses to Key Reviewers' Concerns

### **Technical Rigor & Design Motivation**

- Added all missing technical details (e.g., self-attention formulation, loss function, coefficient optimization procedure).
- Conducted systematic ablation studies validating the necessity of our **multi-token representation** design (**avg. +2.2% gain**) and the **Demo-Interaction** module. Provided t-SNE visualizations as evidence for learned representation expressiveness.
- Added a case study showing how our **importance weighting** dynamically and sensibly allocates attention to demonstrations for a given test query.

### **Experimental Scope & Generalization**

- Expanded evaluation to non-classification tasks. On the text style transfer task (**ParaDetox**), I3CL reduces toxicity to **24.80%**, significantly outperforming all baselines (e.g., ICV: 39.54%), proving its effectiveness beyond simple classification.
- Provided complete multi-model evaluation. We ran full experiments on three backbones: **LLaMA3-8B**, **GLM4-9B**, and **DeepSeek-R1-8B**. I3CL's improvements are consistent and substantial across all, greatly strengthening the reliability of our findings.

### **Efficiency & Scalability Analysis**

- Corrected and extended efficiency analysis. Under a standardized setup (batch size=1), I3CL's inference latency is only **≤1.3%** higher than I2CL's in critical few-shot (5–30 shot) scenarios, with similar memory footprint. Crucially, I3CL avoids the **OOM errors** of standard ICL at large demonstration sizes, proving its scalability.
- The one-time calibration cost (up **34%** at 100-shot) is justified by the substantial accuracy gain (**+11.92%**).

### **Relation to Prior Work & Clarifications**

- Added discussion and citations on **demonstration importance weighting** (e.g., Zhou et al., 2024; Yang et al., 2023) in the Related Work section.
- Explicitly cited the source (**I2CL**) for derived equations.

### **Further Analysis & Robustness**

- Provided a comparative **task vector similarity analysis**, showing that I3CL learns more discriminative task-specific representations (sharper inter-task separation) than I2CL, explaining its superiority from a representation learning perspective.
- Clarified terminology ("weight", "noise") in the text.

---

## 3. Summary of Revisions

- Enhanced methodological details in the main text.
- Added experiments on generative tasks and comprehensive efficiency comparisons.
- Included appropriate citations.
- The appendix contains extensive ablations, the case study, full multi-model results, visualizations, and noise robustness analysis.
- All typos and ambiguous statements pointed out by the reviewers have been corrected.

---

**Conclusion:**
Our revisions have thoroughly and directly addressed all reviewers' concerns, supported by substantial new evidence and analysis. This has strengthened the paper's novelty, rigor, and generalizability. **I3CL presents a new path for implicit in-context learning that balances high efficiency with powerful interaction capabilities**.

Thank you again for your consideration.

**Sincerely,**
The Authors

---

### Meta-Review · Area_Chair_YCTQ · 2026-01-06

**Summary:**

The work studies a lightweight extension of I2CL. I2CL encodes each demonstration independently and directly averages the resulting implicit in-context tokens for in-context learning. In contrast, this work combines inter-demonstration interaction during encoding and demo-to-query interaction during inference via importance weighting.

The main concerns from reviewers (score:6/6/2) for the original version are:

1. Limited backbone coverage (only complete comparison on Llama2-7B), and the evaluated tasks are all classification tasks

2. The overall method seems incremental to I2CL, and the effectiveness of the multi-token design is questioned.

3. The latency comparison is unclear (even incorrect) due to inconsistent or insufficiently specified setups, and the paper also lacks adequate discussion of closely related prior work.

Besides, the AC considers the comparison with LoRA-FT unclear. The supplementary material states “The fine-tuning parameters are listed in Table ??”, but the table is not clearly provided.

**Reviewer Concerns:**

Reviewers generally agreed that the problem is clear and that the improvements are large on the reported classification benchmarks. The rebuttal also improved clarity and completeness by adding missing technical details, expanding experiments to a non-classification generative task (ParaDetox), extending to multiple backbones, providing ablation studies, and correcting the latency comparison.

However, some concerns are only partly addressed in the rebuttal:

1. Incremental novelty (raised strongly by Reviewer 22Y8): Even with the added experiments and analysis, the method may still be incremental relative to I2CL

2. Beyond-classification generality (raised by Reviewers xuZX and 22Y8): Only one non-classification case is tested, and QA-style tasks remain missing.

3. Discussion of related work (raised by Reviewer xuZX): No clear discussion in the rebuttal or the revised manuscript is given for the specific related papers mentioned by this reviewer.

**Reviewer Scores:**

Reviewer X6BH (6): Likely to remain at 6. Their main requests focused on missing technical details and clearer formulation description, which the rebuttal and revision largely address.

Reviewer xuZX (6): Likely to remain at 6. Their concerns on experimental scope beyond classification, missing citations, interpretation of importance weights, and efficiency comparison were mainly addressed.

Reviewer 22Y8 (2, confidence 5): Unlikely to change. The main concern is that the contribution is incremental relative to I2CL, and the rebuttal may not resolve the reviewer’s raw idea.

---

### Decision · Program_Chairs · 2026-01-26

Reject